# Fast Mixture of Curvature-Aware Experts for Diverse and Dynamic Graph Topologies

**Jiayi Yang** [1]  **Xing Wei** [1]  **Chunchun Chen** [2]  **Yi Feng** [1]  **Wengang Guo** [1]  **Rui Fan** [1,2]  **Xiaofeng Cao** [3]  **Xin Sun** [4]  **Wei Ye** [1,2]

## Abstract

Dynamic graph learning, which focuses on modeling the merging, vanishing, and reconnection of nodes and edges, is crucial for real-world applications. In dynamic graphs, node neighborhoods often exhibit diverse and time-evolving topologies, including hierarchical, grid-like, and cyclic patterns. Existing methods typically embed graphs into a single curvature space, which limits the quality of node representations when the embedding geometry is not aligned well with the local graph topology. In this paper, we propose **DyG-MoCE**, a **Dy**namic **G**raph Transformer with a **M**ixture **o**f **C**urvature-aware **E**xperts, which efficiently embeds each node at every timestamp into an adaptive curvature space. Specifically, DyG-MoCE incorporates a mixture-of-experts framework to both the attention and feed-forward modules, where each expert operates on a Riemannian manifold with a distinct curvature. Then, motivated by the geometric continuity across the experts, we introduce a routing mechanism with a ranking constraint. To improve efficiency, we design a fast Riemannian attention module for DyG-MoCE, achieving an average speedup of 27.5% and memory reduction of 52.6%. Notably, the fast Riemannian attention module is broadly applicable to Transformer models with sequence inputs. Extensive experimental results show that DyGMoCE significantly outperforms other state-of-the-art methods.

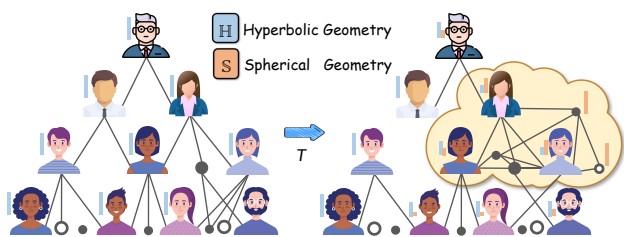

*Figure 1.* Illustration of the embedding geometry adapted to each node. Human icons, circles, and dots denote different nodes. As the graph evolves, the local topology of the same node at different timestamps may correspond to a mixture of embedding geometries, while different nodes may adopt different geometries.

## 1. Introduction

Dynamic graphs are prevalent in numerous real-world scenarios, including social networks (Kumar et al., 2019), traffic networks (Schäfer et al., 2014), and international trade (MacDonald et al., 2015). Such graphs often exhibit diverse local topologies, where grid-like, hierarchical, and cyclic patterns may coexist within a single node neighborhood (Bachmann et al., 2020). These topological patterns are naturally modeled in Euclidean, hyperbolic, and spherical Riemannian geometries, corresponding to different curvatures (Sun et al., 2022). Meanwhile, as nodes and edges emerge, vanish, and reconnect over time, the neighborhood of each node evolves, inducing time-varying structures. Despite this, most existing dynamic graph neural networks (Zou et al., 2024; Peng et al., 2025) embed nodes into the same Euclidean space, which degrades the quality of node representations when the embedding geometry is not aligned well with the local topology.

In recent years, a few dynamic graph neural networks have turned to hyperbolic space (Bai et al., 2023; Li et al., 2024; Xu et al., 2024), motivated by its strong ability to model hierarchical and power-law structures. For example, HG-WaveNet (Bai et al., 2023) employs hyperbolic diffusion and hyperbolic dilated convolutions to capture spatial and temporal dynamics. STGN$^h$ (Xu et al., 2024) adopts a hyperbolic Transformer with an update-gated memory mechanism. However, committing to a single geometry is in-

---

[1]College of Electronic and Information Engineering, Tongji University, Shanghai, China [2]Shanghai Research Institute for Intelligent Autonomous Systems, Tongji University, Shanghai, China [3]School of Computer Science and Technology, Tongji University, Shanghai, China [4]Faculty of Data Science, City University of Macau, Macau, China. Correspondence to: Wei Ye <yew@tongji.edu.cn>.

*Proceedings of the 43$^{rd}$ International Conference on Machine Learning*, Seoul, South Korea. PMLR 306, 2026. Copyright 2026 by the author(s).

adequate for representing diverse and dynamic topologies. While UCG-DG (Sun et al., 2025) allocates nodes to either Euclidean or hyperbolic space, the assignment is non-learnable. Moreover, each node remains confined to a single geometry, precluding effective mixed-curvature representations that adapt to heterogeneous topologies.

Considering the diversity across different substructures and the evolving nature of dynamic graphs as shown in Fig. 1, we aim to enable different nodes, as well as the same node at different timestamps, to adaptively adopt their own appropriate geometry. To achieve this, two main issues must be addressed. The first issue is about *optimal embedding space selection*. The appropriate geometry should take into account not only geometric types such as hyperbolic and spherical, but also the curvature magnitude, ranging from strongly to weakly hyperbolic, Euclidean, and weakly to strongly spherical geometries. The second issue is about *Riemannian computational costs*. Riemannian operations on these geometries, such as exponential and logarithmic mappings, as well as Möbius addition and multiplication, are computationally expensive even on static graphs. Consequently, Riemannian operations must be accelerated to make them available on real-world dynamic graphs, which requires processing interactions at each timestamp.

To this end, we introduce **DyGMoCE**, a **M**ixture **o**f **C**urvature-aware **E**xperts Transformer for **Dy**namic **G**raphs that constructs mixed-curvature embedding spaces for each node at every timestamp. Specifically, we treat curvature-aware Riemannian attention heads and FFNs as experts within the multi-head attention and FFN modules. Then, motivated by the geometric continuity across Riemannian experts with diverse curvatures, we propose a routing mechanism that selects a contiguous block of neighboring experts with the maximized aggregated routing score. Additionally, we regularize routing with a ranking constraint that aligns routed curvatures with empirical neighborhood geometry. For computational efficiency, we reformulate Riemannian attention into a fast module using polar decomposition and the law of cosines in Riemannian spaces. This design yields an average speedup of $27.5\%$ and memory reduction of $52.6\%$ for DyGMoCE. Experimental results on eight benchmark datasets show that DyGMoCE achieves significant improvements in overall performance. Our main contributions are summarized as follows:

- We introduce a mixture of curvature-aware experts in both attention and FFN modules to model diverse and evolving node topologies at each timestamp.

- We propose a routing mechanism to adaptively select geometrically continuous experts.

- We design a fast Riemannian attention module to improve efficiency.

## 2. Related Work

**Dynamic Graph Learning.** Most existing dynamic graph neural networks are built upon GCNs (Xu et al., 2020; Rossi et al., 2020; Cong et al., 2023; Cao et al., 2025b). Recently, Transformers have emerged as a compelling alternative due to their flexibility in modeling temporal events and capturing long-range dependencies. DyGFormer (Yu et al., 2023) introduces neighbor co-occurrence encoding to model node relationships via shared first-order neighbors, and adopts the temporal kernel proposed in TGAT (Xu et al., 2020) for time-interval embedding. To further enhance temporal representations, SimpleDyG (Wu et al., 2024) aligns event sequences within a unified time domain, TIDFormer (Peng et al., 2025) proposes mixed-granularity temporal encoding to capture multiple time scales, and TAMI (Yu et al., 2025) focuses on addressing heterogeneity in temporal interactions. Beyond first-order temporal interactions, several methods explore higher-order patterns in dynamic graphs. RepeatMixer (Zou et al., 2024) integrates higher-order repeat behaviors into neighbor sampling, while TPNet (Lu et al., 2024) employs time-decayed temporal walk matrices. Comprehensive benchmarks (Huang et al., 2023; Yi et al., 2025) have also been established to facilitate systematic evaluation of temporal graph models.

**Riemannian Graph Learning.** Real-world graphs often exhibit topology diversity, necessitating the joint modeling of Euclidean and non-Euclidean GNNs. The $\kappa$-stereographic model (Bachmann et al., 2020) offers a unified framework for Riemannian manifolds with positive, negative, and zero curvature. Building on it, SelfMGNN (Sun et al., 2022) and H-EDML (Cao et al., 2025a) formulate objective functions via contrastive learning and mutual learning. FPS-T (Cho et al., 2023) extends Riemannian learning to Transformers by applying Riemannian linear transformations to value vectors. However, these methods typically assign all nodes to a globally uniform mixed-curvature space. To address this, GeoAwaken (Sun et al., 2024) introduces a distillation approach that assigns each node a Euclidean or hyperbolic teacher according to its Gromov $\delta$-hyperbolicity (Borassi et al., 2015). Recently, mixture-of-experts (MoE) has been introduced to learn adaptive node representations in mixed curvature spaces. In particular, CAT (Lin et al., 2025) adopts an MoE framework to coarsely fuse three graph Transformers with hyperbolic, Euclidean, and spherical geometries, while GraphMoRE (Guo et al., 2025) adaptively selects two-layer Riemannian GCN experts. In contrast, we propose a unified Riemannian Transformer that implements MoE within both the multi-head attention and the FFN modules.

**Riemannian Learning on Dynamic Graphs.** Few studies have explored Riemannian learning in dynamic graphs. Most of them (Bai et al., 2023; Sun et al., 2025) focus on discrete-time dynamic graphs (DTDGs), which represent

the graph as a series of snapshots. For example, DHGAT (Li et al., 2024) embeds nodes in each snapshot into a shared hyperbolic spaces and aggregates their historical representations after attention. In contrast, STGN$^h$ (Xu et al., 2024) targets continuous-time dynamic graphs (CTDGs), which aligns with our setting. However, modeling CTDGs requires processing each interaction, leading to high computational cost in Riemannian spaces. To improve efficiency, we propose the fast Riemannian attention module.

## 3. Preliminary

### 3.1. Problem Formulation

The continuous-time dynamic graph (CTDG) is represented as a sequence of non-decreasing temporal events $\mathcal{G} = \{(u_1, v_1, t_1), (u_2, v_2, t_2), \ldots, (u_I, v_I, t_I)\}$, where $u_i, v_i \in \mathcal{V}$ denote the source and destination nodes of the $i$-th edge at timestamp $t_i$. Here, $\mathcal{V}$ represents the set of nodes in the graph and $I$ is the number of interactions. For each node $v \in \mathcal{V}$, the input to the transformer is constructed by creating an ordered sequence $Seq_v^t$, which collects each node and its historical first-order interactions before time $t$. The node sequence $Seq_v^t$ is then encoded as node embeddings $\mathbf{X} \in \mathbb{R}^{L \times d}$, where $L$ represents the sequence length, and $d$ denotes the embedding dimension. In this paper, we focus on two fundamental tasks in dynamic graph learning: (i) dynamic link prediction, which aims to predict the formation of a link between nodes $u$ and $v$ at a future time $t$, and (ii) dynamic node classification, which predicts the evolving class of $u$ or $v$ at time $t$.

### 3.2. Mixture of Experts

A Mixture of Experts (MoE) module is composed of $M$ experts $\mathcal{E} = \{E_1, E_2, ..., E_M\}$ and a router. In Transformers, each expert is typically implemented as a feed-forward network (FFN). The router learns to assign dynamic weights $[g_1, g_2, ..., g_M]^\top \in \mathbb{R}^M$ to the experts based on $\mathbf{x} \in \mathbb{R}^d$:

$$G(\mathbf{x}) = \text{Softmax}(\text{Linear}(\mathbf{x})). \quad (1)$$

A widely adopted variant is the Top-$k$ router, a sparse routing method that selects the top $k$ experts with the highest weights for each input. The final output $\mathbf{y}$ is computed as:

$$\mathbf{y} = \sum_{i=1}^{M} g_i E_i(\mathbf{x}). \quad (2)$$

In this paper, we leverage the MoE framework to construct mixed-curvature embedding spaces.

### 3.3. Riemannian Attention

The widely adopted Riemannian attention (Xu et al., 2024; Yang et al., 2025) consists of four components:

**Linear Transformation.** Given an input $\mathbf{X} \in \mathbb{R}^{L \times d}$, it is first mapped to the Riemannian manifold via exponential mapping at the origin, then projected by Riemannian linear transformation to obtain the query, key, and value $\mathbf{Q}, \mathbf{K}, \mathbf{V} \in \mathbb{R}^{L \times d}$ representations:

$$\mathbf{Q}, \mathbf{K}, \mathbf{V} = \text{Linear}_{q,k,v}^\kappa(\exp_{\mathbf{0}}^\kappa(\mathbf{X})). \quad (3)$$

**Similarity Score Computation.** The similarity score $\alpha_{ij}$ between node $i$ and $j$ in sequence $Seq_v^t$ is computed using the negative geodesic distance:

$$\alpha_{ij} = -\mathcal{D}_\kappa(\mathbf{Q}_i, \mathbf{K}_j) = -2\tan_\kappa^{-1}\left(\|(-\mathbf{Q}_i) \oplus_\kappa \mathbf{K}_j\|\right). \quad (4)$$

Here, $\tan_\kappa(\mathbf{x})$ denotes the generalized tangent function associated with curvature $\kappa$, defined as $\frac{1}{\sqrt{\kappa}}\tan(\sqrt{\kappa}\|\mathbf{x}\|)$ if $\kappa > 0$, $\frac{1}{\sqrt{-\kappa}}\tanh(\sqrt{-\kappa}\|\mathbf{x}\|)$ if $\kappa < 0$, and $\|\mathbf{x}\|$ if $\kappa = 0$. Its inverse is denoted by $\tan_\kappa^{-1}$. Here, $\mathbf{x}$ is a vector on the Riemannian manifold, and $\kappa > 0$, $\kappa < 0$, and $\kappa = 0$ correspond to spherical, hyperbolic, and Euclidean geometries, respectively. The Möbius addition operator $\oplus_\kappa$ in $\mathbb{R}^d$ is given by:

$$\mathbf{Q}_i \oplus_\kappa \mathbf{K}_j :=$$
$$\frac{(1 + 2\kappa\langle\mathbf{Q}_i, \mathbf{K}_j\rangle + \kappa\|\mathbf{K}_j\|^2)\mathbf{Q}_i + (1 - \kappa\|\mathbf{Q}_i\|^2)\mathbf{K}_j}{1 + 2\kappa\langle\mathbf{Q}_i, \mathbf{K}_j\rangle + \kappa^2\|\mathbf{Q}_i\|^2\|\mathbf{K}_j\|^2}. \quad (5)$$

**Similarity Score Normalization.** The normalized similarity score $w_{ij}$ is calculated across the sequence $Seq_v^t$ using the softmax function.

**Value Aggregation.** To mitigate the computation cost introduced by the sequential nature of Möbius addition, HGAT (Chami et al., 2019) projects the embeddings into Euclidean space, aggregates them, and maps the result back to the Riemannian manifold:

$$\mathbf{H}_i^{\text{att}_\kappa} = \exp_{\mathbf{0}}^\kappa\left(\sum_{j \in Seq_v^t} \log_{\mathbf{0}}^\kappa(w_{ij} \otimes_\kappa \mathbf{V}_j)\right). \quad (6)$$

Here, $\mathbf{H}^{\text{att}_\kappa} \in \mathbb{R}^{L \times d}$ is the Riemannian attention output for node $v$ under the curvature $\kappa$. The logarithmic mapping at the origin is computed as $\log_{\mathbf{0}}^\kappa(\mathbf{x}) = \tan_\kappa^{-1}(\|\mathbf{x}\|)\frac{\mathbf{x}}{\|\mathbf{x}\|}$, and the Möbius scalar multiplication $\otimes_\kappa$ in $\mathbb{R}^d$ is defined as:

$$w_{ij} \otimes_\kappa \mathbf{V}_j := \tan_\kappa\left(w_{ij}\tan_\kappa^{-1}(\|\mathbf{V}_j\|)\right)\frac{\mathbf{V}_j}{\|\mathbf{V}_j\|}. \quad (7)$$

Details on Riemannian geometry are in Appendix D.1.

## 4. DyGMoCE

In this section, we present the details of the proposed DyG-MoCE. As illustrated in Figure 2, a mixture of curvature-aware Riemannian experts is first employed to jointly incorporate diverse expert types and curvature magnitudes in both the attention and FFN modules. Then, a routing

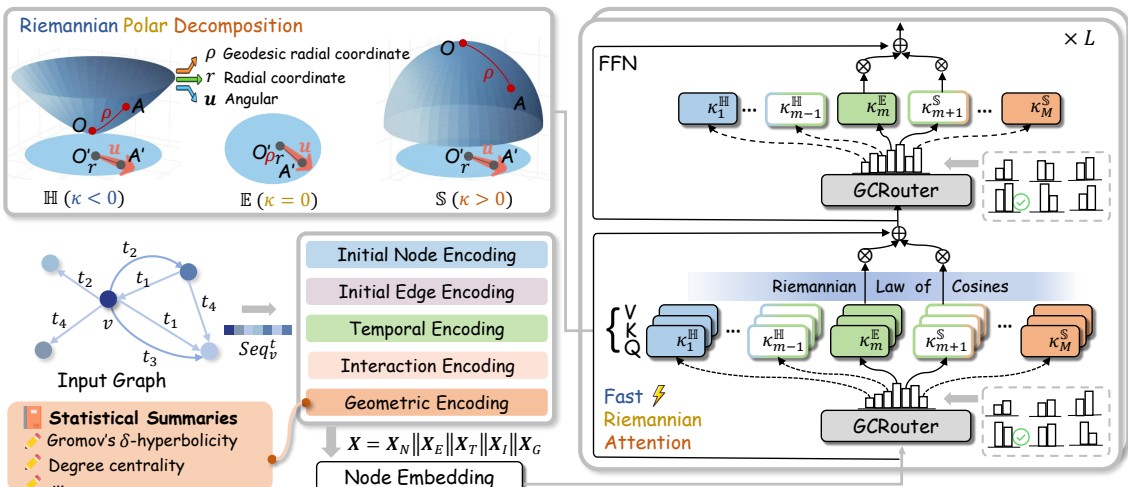

*Figure 2.* The overall framework of DyGMoCE. First, the node embedding is obtained by concatenating the features from node, edge, temporal, interaction, and our geometric encoders. Then, we integrate diverse Riemannian experts in both the attention and FFN modules. The GCRouter with ranking constraints is designed to embed nodes into mixed-curvature spaces. Furthermore, we adopt polar decomposition to obtain the polar components $\rho, r, \mathbf{u}$ of $\mathbf{Q}, \mathbf{K}, \mathbf{V}$, and leverage the Riemannian law of cosines to accelerate the attention.

mechanism with a ranking constraint is proposed to construct an adaptively mixed-curvature space for each node at every timestamp based on its local topology. To improve efficiency, we replace the standard Riemannian attention with a fast alternative. Finally, the node representations are utilized for downstream tasks in dynamic graph learning.

### 4.1. Diverse Curvature-Aware Riemannian Experts

Existing mixed-curvature Riemannian Transformers based on MoE (Lin et al., 2025) coarsely integrate Transformer branches for different geometries, which fails to support joint mixed-curvature modeling in a single Transformer. In contrast, we employ the MoE mechanism in both the multi-head attention and the FFN module, providing a unified architecture for modeling mixed-curvature representations.

**Riemannian Attention Experts.** To accommodate diverse graph topologies, we construct Riemannian experts based on the $\kappa$-stereographic model $\mathcal{M}_\kappa = \left\{ \mathbf{x} \in \mathbb{R}^d \mid -\kappa \|\mathbf{x}\|^2 < 1 \right\}$ (Bachmann et al., 2020), where $\kappa$ denotes the curvature. Positive curvature induces a stereographic sphere model on the spherical manifold $\mathbb{S}$, negative curvature instantiates the Poincaré ball model on the hyperbolic manifold $\mathbb{H}$, and zero curvature corresponds to the Euclidean manifold $\mathbb{E}$. We adopt an MoE perspective on multi-head attention, treating each attention head as a Riemannian expert associated with a distinct curvature. To aggregate these experts, all attention heads are aligned in a shared Euclidean space using the logarithmic mapping at the origin $\log_\mathbf{0}^\kappa$. Since the exponential mapping $\exp_\mathbf{0}^\kappa$ and the logarithmic mapping $\log_\mathbf{0}^\kappa$ form an inverse pair, we preserve the intermediate representations prior to $\exp_\mathbf{0}^\kappa$ in Eq. (6), *i.e.*,

$\mathbf{H}_i^{\text{att}_\kappa} = \sum_{j \in Seq_v^t} \log_\mathbf{0}^\kappa (w_{ij} \otimes_\kappa \mathbf{V}_j)$, where $\mathbf{H}_i^{\text{att}_\kappa}$ denotes the $i$-th row of $\mathbf{H}^{\text{att}_\kappa}$. The Riemannian attention experts are aggregated via a weighted sum:

$$\mathbf{H}^{\text{att}} = \sum_{m=1}^M g_m^{\text{att}} \mathbf{H}^{\text{att}_{\kappa_m}}. \tag{8}$$

Here, $M$ is the number of Riemannian attention experts, and $\kappa_m$ and $g_m^{\text{att}}$ denote the curvature and routing weight for the $m$-th expert. The multi-head attention output $\mathbf{O} \in \mathbb{R}^{L \times d}$ is obtained via a residual connection (He et al., 2016) followed by layer normalization (Ba et al., 2016).

**Riemannian FFN Experts.** Each Riemannian feed-forward (FFN) expert is implemented as a two-layer MLP with curvature $\kappa_m$. In each layer, we employ $\mathcal{F}^{\kappa_m} = \log_\mathbf{0}^{\kappa_m} \circ \text{Linear}^{\kappa_m} \circ \exp_\mathbf{0}^{\kappa_m}$, where $\text{Linear}^{\kappa_m}$ denotes a Riemannian linear transformation with a bias term. Specifically, the input is first mapped to the Riemannian manifold via $\exp_\mathbf{0}^{\kappa_m}$, processed by a Riemannian linear transformation, and mapped back to Euclidean space via $\log_\mathbf{0}^{\kappa_m}$. The Riemannian FFN Expert is as follows:

$$\mathbf{H}^{\text{FFN}_{\kappa_m}} = \mathcal{F}_2^{\kappa_m} (\sigma \, \mathcal{F}_1^{\kappa_m}(\mathbf{O})). \tag{9}$$

Here, $\sigma(\cdot)$ denotes the GELU (Hendrycks, 2016) activation function. The Riemannian FFN experts are combined through a weighted sum:

$$\mathbf{H}^{\text{FFN}} = \sum_{m=1}^M g_m^{\text{FFN}} \mathbf{H}^{\text{FFN}_{\kappa_m}}, \tag{10}$$

where $g_m^{\text{FFN}}$ represents the routing weight for the $m$-th expert. Considering both the curvature type and its magnitude, we assign each expert a distinct initial curvature value. The $M$ geometric experts are sorted by curvature in ascending order

to form the expert set $\mathcal{E} = \{E^{\mathbb{H}}_{\kappa_1}, ..., E^{\mathbb{H}}_{\kappa_{m-1}}, E^{\mathbb{E}}_{\kappa_m}, E^{\mathbb{S}}_{\kappa_{m+1}}, ..., E^{\mathbb{S}}_{\kappa_M}\}$, where the curvatures satisfy $\kappa_1 \leq \kappa_m \leq \kappa_M$.

The input representation $\mathbf{X} = \mathbf{X}_N || \mathbf{X}_E || \mathbf{X}_T || \mathbf{X}_I || \mathbf{X}_G \in \mathbb{R}^{L \times d}$ is constructed by concatenating the features from the node, edge, temporal, interaction encoders (Xu et al., 2020; Yu et al., 2023; Lu et al., 2024), as well as our geometric encoder. To capture higher-order neighborhood patterns, our geometric encoder explicitly incorporates geometric information derived from graph topology. Specifically, we adopt statistical summaries of graph structural properties from prior work (Wang et al., 2025), including degree centrality, betweenness centrality, closeness centrality, clustering coefficient, and square clustering coefficient. Since these structural properties lack an explicit characterization of graph geometry, we introduce two graph geometric properties: Gromov's $\delta$-hyperbolicity and neighborhood growth rate. Graph properties are concatenated and mapped to the geometric feature $\mathbf{X}_G \in \mathbb{R}^{L \times d_G}$ via an MLP. We refer readers to Appendix D.2 for the remaining four encoders and to Appendix D.3 for descriptions of graph properties.

## 4.2. Routing for Geometrically Continuous Experts

Standard MoE architectures use a Top-$k$ router to dynamically select $k$ experts with the highest routing scores, where experts are implemented as independent, functionally identical units. In contrast, DyGMoCE introduces heterogeneous experts with continuously varying Riemannian geometric properties, ranging from strongly to weakly hyperbolic, through Euclidean, to weakly and strongly spherical. Consequently, a node sequence exhibiting a specific local topology is naturally associated with a contiguous block of Riemannian experts, whereas the Top-$k$ router ignores such prior knowledge. To this end, we propose the Geometrically Continuous Expert Router (GCRouter), which employs a sliding window strategy over the ordered expert sequence to select the consecutive $k$ experts with the maximal aggregated routing score.

To simplify the notation, we use $\mathbf{H} \in \mathbb{R}^{L \times d}$ to denote the input to GCRouter in both the attention and FFN modules. The input $\mathbf{X} \in \mathbb{R}^{L \times d}$ is the embedding of the sequence $Seq_v^t$, which is formed by concatenating the center node $v$ with its sampled historical neighbor nodes at time $t$. Since $\mathbf{H}$ is derived from $\mathbf{X}$, the node sequence jointly encodes the local graph topology and thus exhibits consistent geometric properties. Therefore, we perform routing at the sequence level and compute the routing score that is most suitable for each sequence at every timestamp. Specifically, we employ a linear transformation followed by mean pooling to obtain the routing score $\mathbf{s} = \text{Mean}(\text{Linear}(\mathbf{H})) \in \mathbb{R}^M$. Then, we apply the sliding window strategy to segment the ordered expert set $\mathcal{E}$ into a series of overlapping candidate batches $\mathcal{B}$. The $i$-th batch is $\mathcal{B}_i = \{E_{\kappa_j} \mid i \leq j \leq i+k-1, E_{\kappa_j} \in$

$\mathcal{E}, i \in [1, M - k + 1]\}$, with window length $k$ and a stride of 1. The routing scores of the experts within $\mathcal{B}_i$ are denoted by $\mathcal{S}_i = \{\mathbf{s}_i, \mathbf{s}_{i+1}, ..., \mathbf{s}_{i+k-1}\}$. The optimal subset $\hat{\mathcal{S}}$ is defined as the one that maximizes the aggregated routing score $\sum_{\mathbf{s}_i \in \hat{\mathcal{S}}} \mathbf{s}_i$, with $\hat{\mathcal{B}}$ denoting the corresponding expert batch. Finally, we normalize the scores within $\hat{\mathcal{S}}$ to compute gating weights and obtain the output of GCRouter:

$$g_i = \begin{cases} \dfrac{\exp(\mathbf{s}_i)}{\sum_{\mathbf{s}_j \in \hat{\mathcal{S}}} \exp(\mathbf{s}_j)}, & \mathbf{s}_i \in \hat{\mathcal{S}}, \\ 0, & \mathbf{s}_i \notin \hat{\mathcal{S}}. \end{cases} \quad (11)$$

$$\text{GCRouter}(\mathbf{H}) = \sum_{i=1}^M g_i E_{\kappa_i}(\mathbf{H}).$$

**Ranking constraint.** GCRouter is expected to promote routing behaviors that align with the intrinsic geometry encoded in the local neighborhood of each node. Specifically, we define the routed curvature of node $v$ as $\hat{\kappa}_v = \sum_{i=1}^M g_i(v) \kappa_i$, serving as a soft estimate of its geometric properties. Here, $g_i(v)$ denote the gating weight assigned to expert $E_{\kappa_i}$ associated with curvature $\kappa_i$.

Let $\kappa_v^*$ represent the Gromov's $\delta$-hyperbolicity (Borassi et al., 2015) of the local neighborhood of node $v$, quantifying its empirical geometry. For any two nodes $v$ and $u$ satisfying $\kappa_v^* < \kappa_u^*$, we encourage the routed curvature to preserve the same ordering by minimizing the following ranking loss:

$$\mathcal{L}_{\text{rank}} = \sum_{\kappa_v^* < \kappa_u^*} \max(0, \hat{\kappa}_v - \hat{\kappa}_u - \tau). \quad (12)$$

Since Gromov's $\delta$-hyperbolicity quantifies the tree-likeness of a graph rather than exact curvature, a margin hyperparameter $\tau$ is introduced to relax the ordering constraint and enhance robustness.

The overall objective function we aim to minimize consists of the downstream task objectives and the ranking constraints, which can be formulated as:

$$\mathcal{L} = \mathcal{L}_{\text{task}} + \lambda \mathcal{L}_{\text{rank}}, \quad (13)$$

where $\lambda$ is the hyperparameter controlling the trade-off.

## 4.3. Fast Riemannian Attention

Although Riemannian attention has the same time complexity as its Euclidean counterpart, it incurs a substantially higher floating-point operations (FLOPs) cost, which restricts its applicability in real-world scenarios. In this section, we propose a fast Riemannian attention mechanism to improve the efficiency of the Riemannian attention expert. Intuitively, our goal is to mitigate the computational burden by avoiding expensive vector-level operations on Riemannian manifolds, such as the Möbius addition used to compute geodesic distances in Eq. (4) and Möbius scalar multiplication employed during value aggregation in Eq. (7).

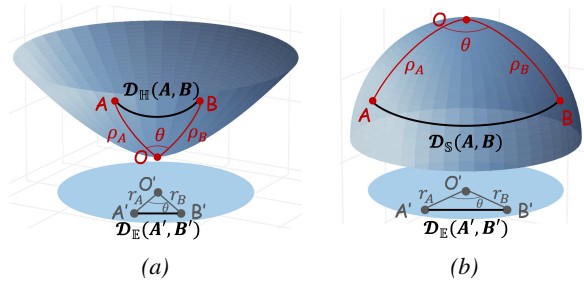

*Figure 3.* Illustration of the Riemannian law of cosines under hyperbolic and spherical geometries. (a) Hyperbolic law of cosines in the Poincaré ball model ($\kappa < 0$). (b) Spherical law of cosines in the stereographic sphere model ($\kappa > 0$).

To achieve this, we exploit the polar coordinates in Riemannian geometry, reformulating geodesic distances using the Riemannian law of cosines and simplifying the value aggregation process.

Considering a nonzero vector $\mathbf{x} \in \mathbb{R}^d \setminus \{\mathbf{0}\}$ on the manifold $\mathcal{M}_\kappa = \{\mathbf{x} \mid -\kappa\|\mathbf{x}\|^2 < 1\}$, its Euclidean polar coordinates are expressed as:

$$\mathbf{x} = r\,\mathbf{u}, \quad r = \|\mathbf{x}\|, \quad \mathbf{u} = \frac{\mathbf{x}}{\|\mathbf{x}\|}. \tag{14}$$

Here, $r \in \mathbb{R}_+$ represents the Euclidean radial coordinate, and $\mathbf{u}$ specifies the angular direction on the unit sphere $\mathbb{S}^{d-1}$. Since the Riemannian metric is conformal to the Euclidean metric, *i.e.*, $g_\mathbf{x}^\kappa = \left(\lambda_\mathbf{x}^\kappa\right)^2 \mathbf{I}$ where $\lambda_\mathbf{x}^\kappa = 2/(1 + \kappa\|\mathbf{x}\|^2)$, the angular component of Riemannian geometry coincides with the Euclidean direction $\mathbf{u}$. Additionally, the corresponding geodesic radial coordinate $\rho = \mathcal{D}_\kappa(\mathbf{0}, \mathbf{x}) = 2\tan_\kappa^{-1}(r) \in \mathbb{R}_+$ is determined by the geodesic distance from the origin $\mathbf{0}$ to $\mathbf{x}$. Detailed derivations of Riemannian polar coordinates are listed in Appendix E.1. For the query, key, and value representations $\mathbf{Q}$, $\mathbf{K}$, and $\mathbf{V}$ in the Riemannian linear transformations as defined in Eq. (3), we compute their polar components $r^\mathbf{Q}, \rho^\mathbf{Q}, r^\mathbf{K}, \rho^\mathbf{K}, r^\mathbf{V}, \rho^\mathbf{V} \in \mathbb{R}^L$, and $\mathbf{u}^\mathbf{Q}, \mathbf{u}^\mathbf{K}, \mathbf{u}^\mathbf{V} \in \mathbb{R}^{L \times d}$, respectively.

To speed up computation of similarity scores in Eq. (4), we reformulate the geodesic distance $\mathcal{D}_\kappa(\mathbf{Q}_i, \mathbf{K}_j)$ using the Riemannian law of cosines. As illustrated in Figure 3, the lengths of two sides $OA$ and $OB$ correspond to the geodesic radial coordinate $\rho_i^\mathbf{Q}$ and $\rho_j^\mathbf{K}$, while $O'A'$ and $O'B'$ represent the Euclidean radial coordinate $r_i^\mathbf{Q}$ and $r_j^\mathbf{K}$. The angle $\theta$ between the two sides $OA$ and $OB$, as well as $O'A'$ and $O'B'$, are determined by the inner product of unit vector $\mathbf{u}_i^\mathbf{Q}$ and $\mathbf{u}_j^\mathbf{K}$, yielding $\cos\theta_{ij} = \mathbf{u}_i^\mathbf{Q} \cdot (\mathbf{u}_j^\mathbf{K})^T$. By applying the Riemannian law of cosines, the geodesic distance along the third side can be computed from the lengths of the other two sides and their included angle. Therefore, the target geodesic distance $\mathcal{D}_\kappa(\mathbf{Q}_i, \mathbf{K}_j)$ corresponding to side $AB$ is

reformulated as:

$$\begin{aligned}
\mathcal{D}_\kappa(\mathbf{Q}_i, \mathbf{K}_j) = \cos_\kappa^{-1}\big(&\cos_\kappa(\rho_i^\mathbf{Q})\cos_\kappa(\rho_j^\mathbf{K}) \\
&+ \operatorname{sign}_\kappa \sin_\kappa(\rho_i^\mathbf{Q})\sin_\kappa(\rho_j^\mathbf{K})\, \cos\theta_{ij}\big),
\end{aligned} \tag{15}$$

where $\sin_\kappa(\cdot)$ and $\cos_\kappa(\cdot)$ denote the generalized trigonometric functions associated with curvature $\kappa$. Specifically, for $\kappa > 0$, these are defined as $\sin_\kappa(x) = \sin(\sqrt{\kappa}x)$ and $\cos_\kappa(x) = \cos(\sqrt{\kappa}x)$, whereas for $\kappa < 0$, they are given by $\sin_\kappa(x) = \sinh(\sqrt{-\kappa}x)$ and $\cos_\kappa(x) = \cosh(\sqrt{-\kappa}x)$. The function $\operatorname{sign}_\kappa$ takes the value 1 for $\kappa > 0$ and $-1$ for $\kappa < 0$. As $\kappa \to 0$, the distance $\mathcal{D}_\kappa(\mathbf{Q}_i, \mathbf{K}_j)$ smoothly converges to the Euclidean formula: $\mathcal{D}_0(\mathbf{Q}_i, \mathbf{K}_j) = 2\|\mathbf{Q}_i - \mathbf{K}_j\|$.

For value aggregation, the polar decomposition enables us to combine Möbius scalar multiplication and the logarithmic mapping into a remarkably simple multiplication operation:

$$\begin{aligned}
\mathbf{H}_i^{\mathrm{att}_\kappa} &= \sum\nolimits_{j \in Seq_v^t} \log_\mathbf{0}^\kappa\big(w_{ij} \otimes_\kappa \mathbf{V}_j\big) \\
&= \sum\nolimits_{j \in Seq_v^t} \log_\mathbf{0}^\kappa\big(\tan_\kappa\big(w_{ij}\tan_\kappa^{-1}(r_j^\mathbf{V})\big)\,\mathbf{u}_j^\mathbf{V}\big) \\
&= \sum\nolimits_{j \in Seq_v^t} \log_\mathbf{0}^\kappa\big([\tan_\kappa\big(\tfrac{1}{2}\,w_{ij}\,\rho_j^\mathbf{V}\big)]\,\mathbf{u}_j^\mathbf{V}\big) \\
&= \tfrac{1}{2} \sum\nolimits_{j \in Seq_v^t} w_{ij}\,\rho_j^\mathbf{V}\,\mathbf{u}_j^\mathbf{V}.
\end{aligned} \tag{16}$$

Our method significantly reduces the computational cost of computing similarity scores and performing value aggregation. Details of the Riemannian law of cosines and FLOPs calculations are provided in Appendices E.2 and F.

## 5. Experiments

### 5.1. Experimental Settings

**Datasets and Baselines.** We conduct experiments on eight datasets (Yu et al., 2023) spanning diverse domains, including Wikipedia, Reddit, MOOC, LastFM, Enron, US Legis., UN Trade, and UN Vote. To assess the effectiveness of our proposed model, DyGMoCE, we compare it against eight state-of-the-art baseline methods. These baselines include seven continuous-time dynamic graph neural networks (CT-DGs): TGAT (Xu et al., 2020), GraphMixer (Cong et al., 2023), DyGFormer (Yu et al., 2023), RepeatMixer (Zou et al., 2024), TPNet (Lu et al., 2024), TIDFormer (Peng et al., 2025), and TAMI (Yu et al., 2025), as well as a hyperbolic CTDG, STGN[h] (Xu et al., 2024). Dataset details are provided in Appendix G.

**Evaluation Metrics and Implementation Details.** We evaluate DyGMoCE on downstream tasks including dynamic link prediction and node classification. For link prediction, we report Average Precision (AP) and AUC-ROC under both transductive and inductive settings. Following prior work (Poursafaei et al., 2022; Peng et al., 2025), we adopt three negative sampling strategies, *i.e.*, random (rnd), histor-

*Table 1.* AP for transductive dynamic link prediction with random, historical, and inductive negative sampling strategies (NSS). The best and runner-up results are emphasized by **bold** and underlined fonts. Rank indicates the average rank across datasets.

| NSS | Methods | Wikipedia | Reddit | MOOC | LastFM | Enron | US Legis. | UN Trade | UN Vote | Rank |
|---|---|---|---|---|---|---|---|---|---|---|
| rnd | TGAT | $96.94_{\pm0.06}$ | $98.63_{\pm0.06}$ | $85.84_{\pm0.15}$ | $73.42_{\pm0.21}$ | $71.12_{\pm0.97}$ | $68.52_{\pm3.16}$ | $61.47_{\pm0.18}$ | $52.21_{\pm0.98}$ | 8.25 |
| | GraphMixer | $97.25_{\pm0.03}$ | $97.31_{\pm0.01}$ | $82.78_{\pm0.15}$ | $75.61_{\pm0.24}$ | $82.25_{\pm0.16}$ | $70.74_{\pm1.02}$ | $62.61_{\pm0.27}$ | $52.11_{\pm0.16}$ | 7.88 |
| | DyGFormer | $99.03_{\pm0.02}$ | $99.22_{\pm0.01}$ | $87.52_{\pm0.49}$ | $93.00_{\pm0.12}$ | $92.47_{\pm0.12}$ | $71.11_{\pm0.59}$ | $66.46_{\pm1.29}$ | $55.55_{\pm0.42}$ | 5.13 |
| | RepeatMixer | $99.12_{\pm0.03}$ | $99.07_{\pm0.01}$ | $92.94_{\pm0.08}$ | $91.42_{\pm0.10}$ | $93.89_{\pm0.11}$ | $68.67_{\pm0.36}$ | $63.71_{\pm0.15}$ | $55.32_{\pm0.02}$ | 5.38 |
| | TPNet | $99.32_{\pm0.03}$ | $99.27_{\pm0.00}$ | $94.15_{\pm0.12}$ | $94.50_{\pm0.08}$ | $92.29_{\pm0.15}$ | $80.27_{\pm0.02}$ | $86.37_{\pm0.67}$ | $75.16_{\pm0.05}$ | 2.50 |
| | STGN$^h$ | $99.30_{\pm0.03}$ | $99.25_{\pm0.01}$ | $90.81_{\pm0.16}$ | $89.52_{\pm0.04}$ | $92.16_{\pm0.06}$ | $75.21_{\pm0.46}$ | $69.68_{\pm0.29}$ | $63.54_{\pm1.35}$ | 4.25 |
| | TIDFormer | $99.27_{\pm0.02}$ | $99.17_{\pm0.11}$ | $91.91_{\pm0.17}$ | $93.73_{\pm0.93}$ | $91.57_{\pm0.23}$ | $68.40_{\pm3.78}$ | $50.88_{\pm1.97}$ | $59.49_{\pm0.92}$ | 5.63 |
| | TAMI | $99.27_{\pm0.03}$ | $99.29_{\pm0.01}$ | $87.61_{\pm0.29}$ | $92.63_{\pm0.01}$ | $92.96_{\pm0.27}$ | $67.09_{\pm0.39}$ | $64.86_{\pm0.21}$ | $58.28_{\pm2.11}$ | 5.00 |
| | DyGMoCE | **$99.38_{\pm0.02}$** | **$99.32_{\pm0.01}$** | **$96.83_{\pm0.18}$** | **$95.47_{\pm0.07}$** | **$94.55_{\pm0.15}$** | **$84.43_{\pm0.33}$** | **$90.20_{\pm0.23}$** | **$78.24_{\pm0.08}$** | **1.00** |
| hist | TGAT | $87.38_{\pm0.22}$ | $79.55_{\pm0.20}$ | $82.19_{\pm0.62}$ | $71.59_{\pm0.24}$ | $64.07_{\pm1.05}$ | $62.14_{\pm6.60}$ | $55.74_{\pm0.91}$ | $52.96_{\pm2.14}$ | 7.50 |
| | GraphMixer | $90.90_{\pm0.10}$ | $78.44_{\pm0.18}$ | $77.77_{\pm0.92}$ | $72.47_{\pm0.49}$ | $75.63_{\pm0.73}$ | $81.65_{\pm1.02}$ | $57.05_{\pm1.22}$ | $51.20_{\pm1.60}$ | 7.25 |
| | DyGFormer | $82.23_{\pm2.54}$ | $81.57_{\pm0.67}$ | $85.85_{\pm0.66}$ | $81.57_{\pm0.48}$ | $77.98_{\pm0.92}$ | $85.30_{\pm3.88}$ | $64.41_{\pm1.40}$ | $60.84_{\pm1.58}$ | 5.50 |
| | RepeatMixer | $91.36_{\pm0.43}$ | $84.72_{\pm0.78}$ | $88.19_{\pm0.58}$ | $86.73_{\pm0.34}$ | **$87.38_{\pm0.18}$** | $81.44_{\pm1.07}$ | $65.51_{\pm15.71}$ | $58.70_{\pm0.61}$ | 3.63 |
| | TPNet | $83.84_{\pm0.04}$ | $80.06_{\pm0.17}$ | $90.51_{\pm1.46}$ | $88.01_{\pm0.93}$ | $80.01_{\pm0.49}$ | $93.55_{\pm1.45}$ | $85.28_{\pm1.22}$ | $72.09_{\pm0.29}$ | 3.38 |
| | STGN$^h$ | $90.96_{\pm0.36}$ | $82.06_{\pm0.26}$ | $88.74_{\pm1.26}$ | $82.33_{\pm0.41}$ | $79.06_{\pm1.29}$ | $87.75_{\pm3.32}$ | $69.74_{\pm2.06}$ | $74.09_{\pm0.27}$ | 3.25 |
| | TIDFormer | $83.25_{\pm2.36}$ | $80.50_{\pm1.07}$ | $86.83_{\pm2.00}$ | $86.52_{\pm0.86}$ | $77.37_{\pm1.62}$ | $58.96_{\pm5.37}$ | $49.70_{\pm0.67}$ | $45.25_{\pm0.98}$ | 7.00 |
| | TAMI | $76.71_{\pm8.77}$ | $81.49_{\pm0.63}$ | $80.91_{\pm3.25}$ | $78.46_{\pm0.60}$ | $78.17_{\pm1.34}$ | $84.62_{\pm0.26}$ | $53.14_{\pm0.22}$ | $62.80_{\pm8.98}$ | 6.38 |
| | DyGMoCE | **$91.76_{\pm0.21}$** | **$85.62_{\pm1.94}$** | **$91.37_{\pm1.76}$** | **$88.94_{\pm0.74}$** | $84.21_{\pm0.29}$ | **$96.34_{\pm1.31}$** | **$88.49_{\pm1.64}$** | **$79.36_{\pm0.65}$** | **1.13** |
| ind | TGAT | $87.00_{\pm0.16}$ | $89.59_{\pm0.24}$ | $75.95_{\pm0.64}$ | $71.13_{\pm0.17}$ | $63.94_{\pm1.36}$ | $61.91_{\pm5.82}$ | $60.61_{\pm1.24}$ | $52.89_{\pm1.61}$ | 6.38 |
| | GraphMixer | $88.59_{\pm0.17}$ | $85.26_{\pm0.11}$ | $74.27_{\pm0.92}$ | $68.12_{\pm0.33}$ | $75.01_{\pm0.79}$ | $79.63_{\pm0.84}$ | $60.15_{\pm1.29}$ | $51.60_{\pm0.73}$ | 7.00 |
| | DyGFormer | $78.29_{\pm5.38}$ | $91.11_{\pm0.40}$ | $81.24_{\pm0.69}$ | $73.97_{\pm0.50}$ | $77.41_{\pm0.89}$ | $81.25_{\pm3.62}$ | $55.79_{\pm1.02}$ | $51.91_{\pm0.84}$ | 5.88 |
| | RepeatMixer | **$90.24_{\pm0.36}$** | $92.37_{\pm0.25}$ | $83.11_{\pm1.28}$ | $75.46_{\pm0.78}$ | $83.17_{\pm0.50}$ | $80.21_{\pm0.23}$ | $57.79_{\pm11.3}$ | $50.88_{\pm1.06}$ | 4.25 |
| | TPNet | $83.03_{\pm0.20}$ | $88.57_{\pm0.20}$ | $85.26_{\pm2.44}$ | $77.45_{\pm2.12}$ | $78.44_{\pm1.52}$ | $91.02_{\pm1.21}$ | $87.34_{\pm1.10}$ | $74.36_{\pm0.45}$ | 3.75 |
| | STGN$^h$ | $86.92_{\pm0.24}$ | $91.16_{\pm0.36}$ | $80.41_{\pm0.86}$ | $72.89_{\pm1.12}$ | $78.69_{\pm0.54}$ | $86.02_{\pm0.72}$ | $68.84_{\pm2.69}$ | $69.13_{\pm0.52}$ | 4.25 |
| | TIDFormer | $85.78_{\pm0.84}$ | $92.78_{\pm0.51}$ | $84.53_{\pm1.11}$ | $75.69_{\pm1.17}$ | $79.67_{\pm0.66}$ | $55.30_{\pm4.29}$ | $49.57_{\pm0.96}$ | $42.41_{\pm1.12}$ | 5.50 |
| | TAMI | $63.88_{\pm8.87}$ | $89.58_{\pm1.03}$ | $76.10_{\pm1.65}$ | $67.74_{\pm0.95}$ | $75.53_{\pm0.36}$ | $81.75_{\pm1.39}$ | $53.69_{\pm1.25}$ | $67.98_{\pm1.73}$ | 6.88 |
| | DyGMoCE | $89.15_{\pm0.34}$ | **$92.94_{\pm1.87}$** | **$86.02_{\pm2.31}$** | **$81.48_{\pm1.61}$** | **$84.96_{\pm0.69}$** | **$92.65_{\pm0.91}$** | **$89.76_{\pm1.14}$** | **$78.96_{\pm0.41}$** | **1.13** |

ical (hist), and inductive (ind). For node classification, we use AUC-ROC as the evaluation metric.

DyGMoCE adopts a two-layer Transformer backbone. The number of attention heads equals the number of curvature-aware experts, which is kept the same in the Fast Riemannian attention and FFN modules. Time, interaction, and geometric feature dimensions are all set to 50. We tune the margin hyperparameter $\tau$ and the trade-off coefficient $\lambda$ over $\{0.1, 0.3, 0.5, 1\}$, the dropout rate over $\{0.0, 0.1, 0.2, 0.3, 0.4, 0.5\}$, the number of sampled first-order neighbors over $\{16, 32, 64\}$, the learning rate over $\{10^{-3}, 10^{-4}, 10^{-5}, 10^{-6}\}$, and the weight decay over $\{0.0, 10^{-6}, 10^{-4}\}$. All models are trained for up to 100 epochs with early stopping. For both tasks, we split each dataset chronologically into 70%/15%/15% for training/validation/testing, following (Yu et al., 2023; Lu et al., 2024). Baselines are evaluated using their official implementations and original protocols. Experiments run on an Ubuntu server with an Intel Xeon Gold 6330 CPU and six NVIDIA GeForce RTX 3090 GPUs. For fair comparison, all runtime measurements in Table 3 and Table 10 are conducted on a single GPU. For each experiment, we conduct five independent runs and report the average performance. Our code is available at `https://github.com/YJYTJ/DyGMoCE`.

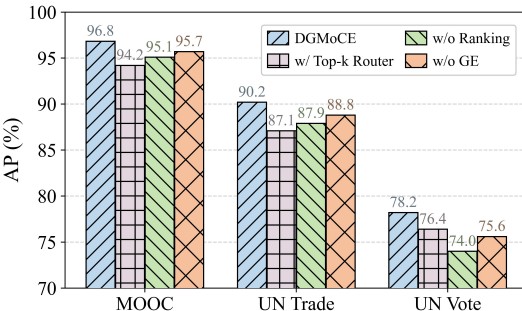

*Figure 4.* Ablation study of DyGMoCE. We compare the DyG-MoCE model (GCRouter + Ranking + GE) with variants: w/ Top-$k$ router, w/o ranking, and w/o geometric encoding (GE).

### 5.2. Experimental Results

The performance of DyGMoCE and other SOTA baselines on the AP metric for transductive dynamic link prediction is shown in Table 1. Results for AUC-ROC in the transductive setting, as well as AP and AUC-ROC under the inductive setting, are provided in Appendix H.1. DyGMoCE achieves the best performance among all the baselines on most of the datasets, with average ranks of 1.00, 1.13, and 1.13 for random, historical, and inductive negative sampling strategies, respectively. Specifically, DyGMoCE significantly outperforms the second-best method on US Legis., UN Trade, and

*Table 2.* AUC-ROC for dynamic node classification.

| Methods | Wikipedia | Reddit | Rank |
|---|---|---|---|
| TGAT | $84.09_{\pm1.27}$ | $70.04_{\pm1.09}$ | 5.5 |
| GraphMixer | $86.80_{\pm0.79}$ | $64.22_{\pm3.32}$ | 7.5 |
| DyGFormer | $87.44_{\pm1.08}$ | $68.00_{\pm1.74}$ | 6.0 |
| RepeatMixer | $88.26_{\pm1.45}$ | $69.74_{\pm1.21}$ | 2.5 |
| TPNet | $87.42_{\pm0.62}$ | $69.24_{\pm1.39}$ | 5.0 |
| STGN$^h$ | $86.43_{\pm1.55}$ | $68.07_{\pm0.94}$ | 7.0 |
| TIDFormer | $87.53_{\pm1.12}$ | $69.59_{\pm1.70}$ | 3.5 |
| TAMI | $85.84_{\pm0.58}$ | $68.47_{\pm1.54}$ | 7.0 |
| DyGMoCE | $\mathbf{89.19_{\pm0.62}}$ | $\mathbf{70.58_{\pm1.21}}$ | 1.0 |

*Table 3.* Training time and memory usage of DyGMoCE with Euclidean attention (EA), Riemannian attention (RA), and our fast Riemannian attention (Fast RA). Reduction ratios (↓%) show the relative time and memory savings of fast Riemannian attention over Riemannian attention. $L$ is the input length. OOM stands for Out-Of-Memory.

| Datasets | $L$ | Running Time EA | RA | Fast RA | ↓ | Memory Usage EA | RA | Fast RA | ↓ |
|---|---|---|---|---|---|---|---|---|---|
| MOOC | 16 | 5 min 26 s | 9 min 03 s | 7 min 27 s | 17.7% | 3,016 MB | 7,599 MB | 4,949 MB | 34.8% |
| | 32 | 6 min 12 s | 12 min 28 s | 9 min 02 s | 27.5% | 5,591 MB | 20,253 MB | 9,530 MB | 52.9% |
| | 64 | 8 min 36 s | — | 14 min 30 s | — | 10,726 MB | OOM | 18,627 MB | — |
| UN Vote | 16 | 15 min 38 s | 27 min 52 s | 18 min 49 s | 32.5% | 3,442 MB | 8,043 MB | 5,395 MB | 32.9% |
| | 32 | 17 min 51 s | 36 min 35 s | 24 min 07 s | 34.1% | 6027 MB | 20,764 MB | 9,989 MB | 50.9% |
| | 64 | 21 min 27 s | — | 40 min 11 s | — | 1,1149 MB | OOM | 19,076 MB | — |

UN Vote datasets under random sampling, achieving improvements of $5.2\%$, $4.4\%$, and $4.1\%$, respectively. Among the baselines, the hyperbolic model STGN$^h$ ranks second under historical sampling, while Euclidean-based method TPNet achieves the second-best performance under random and inductive samplings. However, their performance varies across sampling strategies, indicating that relying on a single geometry is insufficient to capture the diverse topologies in dynamic graphs. In contrast, DyGMoCE leverages a mixture of curvature-aware experts to embed node representations in an adaptive space, exhibiting consistently strong performance across different sampling strategies. As shown in Table 2, DyGMoCE achieves the best average rank on both datasets for the dynamic node classification task, further demonstrating the effectiveness of our approach.

### 5.3. Ablation Study

To further validate the effectiveness of each component in DyGMoCE, we conduct ablation studies with three variants: 1) "*w/ Top-k Router*" replaces the proposed GCRouter and its ranking constraint with a standard Top-$k$ router; 2) "*w/o Ranking*" removes the ranking constraint from the GCRouter; and 3) "*w/o GE*" removes the geometric encoding. We perform ablations on MOOC, UN Trade, and UN Vote, which exhibit distinct Gromov's $\delta$-hyperbolicity over time. Specifically, the average node hyperbolicity ranges from 0.019 to 1.850 on MOOC and from 0.092 to 1.896 on UN Trade, while UN Vote exhibits a smaller range from 0 to 0.627. As shown in Figure 4, removing any component degrades performance, while their combination boosts it. In particular, replacing the GCRouter and its ranking constraint with a Top-$k$ router leads to inferior performance, indicating that our design achieves more effective curvature-aware expert selection by accounting for geometric continuity across Riemannian experts. For the UN Vote dataset, removing either the ranking constraint or geometric encoding results in a substantial performance drop. This suggests that when hyperbolicity variation is limited, explicit geometric information is essential for providing discriminative routing signals and guiding expert assignment.

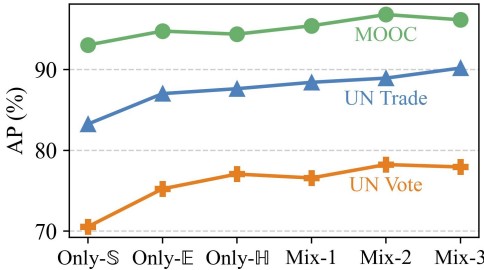

*Figure 5.* Performance of different expert combinations.

### 5.4. Efficiency Analysis

We evaluate the computational efficiency of DyGMoCE by comparing the training time per epoch and GPU memory usage when equipped with conventional Riemannian attention (RA) and our fast Riemannian attention (fast RA). For a fair and comprehensive comparison, we also report the results of DyGMoCE using Euclidean attention (EA). As shown in Table 3, fast RA consistently improves efficiency over conventional RA on both MOOC and UN Vote datasets across different input lengths $L$. Results on additional datasets are reported in Table 10 in Appendix H.2. At $L = 32$, fast RA reduces the training time by $27.5\%$ and decreases memory consumption by $52.6\%$ compared to RA across all the datasets. Moreover, the reduction ratios increase from $L = 16$ to $L = 32$, indicating better scalability with longer input sequences. Notably, RA runs into out-of-memory at $L = 64$ on both datasets, whereas fast RA remains feasible, enabling training with longer historical sequences. Although fast RA remains more computationally demanding compared to EA, it substantially narrows the gap while preserving the accuracy benefits of Riemannian learning, making the trade-off favorable in practice.

### 5.5. Expert Combination Analysis

We evaluate curvature-aware expert combinations in MOOC, UN Trade, and UN Vote datasets. Following (Guo et al., 2025), curvature values are selected from $\{-3, -2, -1, 0, 1, 2, 3\}$ to span multiple magnitudes while maintaining nu-

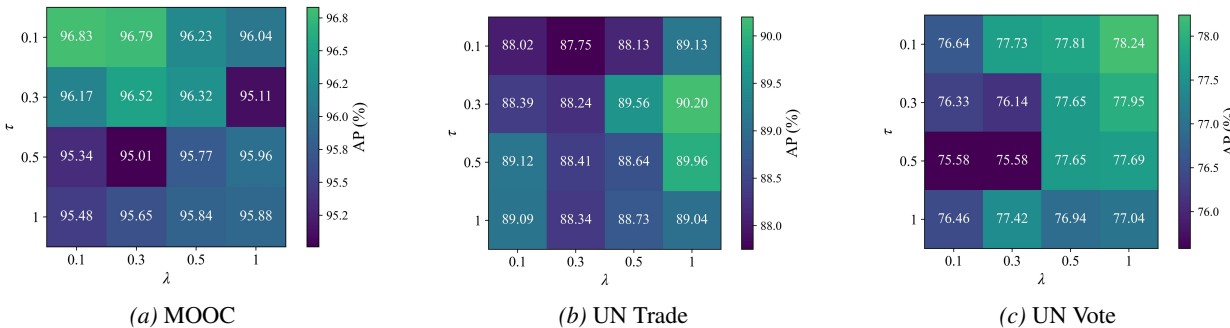

*Figure 6.* Hyperparameter sensitivity of margin $\tau$ in ranking loss and trade-off $\lambda$ in the overall objective.

merical stability. We consider six expert configurations, including three single-geometry variants and three mixed-geometry variants that combine hyperbolic ($\mathbb{H}$), Euclidean ($\mathbb{E}$), and spherical ($\mathbb{S}$) experts. The single-geometry variants include only-$\mathbb{S} = \{1, 2, 3\}$, only-$\mathbb{E} = \{0, 0, 0\}$, and only-$\mathbb{H} = \{-3, -2, -1\}$. The mixed-geometry variants are Mix-1 $=\{-1, 0, 1\}$, Mix-2 $= \{-2, -1, 0, 1, 2\}$, and Mix-3 $= \{-3, -2, -1, 0, 1, 2, 3\}$. As shown in Figure 5, configurations that combine all three geometric types consistently outperform those using a single geometry across all datasets. Moreover, increasing the diversity of curvatures generally leads to performance gains. These results indicate that DyGMoCE leverages complementary properties of hyperbolic, Euclidean, and spherical geometries, and that richer curvature diversity further strengthens this effect.

### 5.6. Sensitivity Analysis

We study the sensitivity of DyGMoCE to two hyperparameters: the margin $\tau$ in the ranking loss of Eq. (12) and the trade-off coefficient $\lambda$ in the overall objective of Eq. (13). As shown in Figure 6, DyGMoCE is relatively stable across a broad range of hyperparameters. On the MOOC dataset, the best performance is obtained with relatively small $\tau$ and $\lambda$. On UN Trade and UN Vote datasets, the best performance is achieved when $\lambda = 1$, while varying $\tau$ causes minor changes. Overall, DyGMoCE remains robust to different hyperparameter choices.

## 6. Conclusion

In this paper, we propose DyGMoCE, a mixture of curvature-aware experts model that adapts embedding geometry to diverse and evolving local topologies in dynamic graphs. DyGMoCE incorporates curvature-aware experts into both the attention and feed-forward modules. We further introduce a routing mechanism over geometrically continuous experts to realize a mixed-curvature representation. In addition, we develop a fast Riemannian attention mechanism to improve efficiency. In the future, we will extend our method to graph foundation models.

## Impact Statement

This paper presents work whose goal is to advance the field of machine learning. There are many potential societal consequences of our work, none of which we feel must be specifically highlighted here.

## Acknowledgments

We thank the anonymous reviewers for their valuable and constructive comments. This work was partially supported by the National Natural Science Foundation of China under Grants 62576253, 62176184, 62476109, AI for Science Program, Shanghai Municipal Commission of Economy and Informatization under Grant 2025-GLZ-RGZN-BTBX-02016, the Fundamental Research Funds for the Central Universities, and the Science and Technology Development Fund, Macao SAR No. 0006/2024/RIA1.

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

## A. Limitations and Future Work

One potential limitation of DyGMoCE lies in the information redundancy among mixed-curvature experts. Although hyperbolic, Euclidean, and spherical experts are each tailored to capture unique topologies, they inevitably encode overlapping, generic knowledge. In conventional mixture-of-experts (MoE) architectures, shared experts are often introduced to explicitly model such common patterns, improving parameter efficiency and expert specialization. However, directly designating a curvature-aware expert as a shared expert is unsuitable, as each expert operates on a distinct manifold with different computational primitives. While our geometrically continuous routing mechanism with ranking constraints partially mitigates this issue by routing nodes according to local hyperbolicity, redundancy remains an open challenge. In the future, we plan to explore knowledge distillation techniques to extract shared consensus across experts, allowing each expert to better focus on its curvature-specific knowledge.

## B. Pseudocode

Algorithm 1 summarizes the overall training pipeline of DyGMoCE.

---

**Algorithm 1:** DyGMoCE

---

**Input:** Continuous-time dynamic graph $\mathcal{G} = \{(u_i, v_i, t_i)\}_{i=1}^{I}$, curvature set $\{\kappa_1, \ldots, \kappa_M\}$, statistical summaries of
graph properties, sequence length $L$, number of experts $k$, number of Transformer layers $L$, trade-off coefficient
$\lambda$, ranking margin $\tau$
**Output:** $\mathbf{H}^{(L)}$ for downstream tasks

1 **foreach** $v \in \{u_i, v_i\}_{i=1}^{I}$ **do**
2      Construct the ordered sequence $Seq_v^t$ for node $v$;
3      Generate node, edge, temporal, interaction, and graph property features $X_N, X_E, X_T, X_I, X_G$;
4      Arrange the features according to $Seq_v^t$;
5      $\mathbf{X} \leftarrow \mathbf{X}_N || \mathbf{X}_E || \mathbf{X}_T || \mathbf{X}_I || \mathbf{X}_G \in \mathbb{R}^{L \times d}$;
6 Initialize model parameters $\Theta$;
7 $\mathbf{H}^{(0)} \leftarrow \mathbf{X}$;
8 **for** $\ell = 1$ **to** $L$ **do**
     // Step1: Mixture of Fast Riemannian Attention Experts
9      **for** $m = 1$ **to** $M$ **do**
10          Compute the fast Riemannian attention representation $\mathbf{H}^{\text{att}\kappa_m}$ for each expert $E_{\kappa_m}^{\text{att}}$ using Eq. (14)-Eq. (16);
11      Compute the expert weights $g^{\text{att}}$ using GCRouter in Eq. (11);
12      Compute the ranking loss $\mathcal{L}_{\text{rank}}^{\text{att}}$ in Eq. (12);
13      Aggregate the attention experts into $\mathbf{H}^{\text{att}}$ via a weighted sum as in Eq. (8);
14      $\mathbf{O}^{(l)} \leftarrow Norm(\mathbf{H}^{\text{att}} + \mathbf{H}^{(l-1)})$;
     // Step2: Mixture of Riemannian Feed-Forward Network Experts
15      **for** $m = 1$ **to** $M$ **do**
16          Compute the Riemannian FFN $\mathbf{H}^{\text{FFN}\kappa_m}$ for each expert $E_{\kappa_m}^{\text{FFN}}$ using Eq. (9);
17      Compute the expert weights $g^{\text{FFN}}$ using GCRouter in Eq. (11);
18      Compute the ranking loss $\mathcal{L}_{\text{rank}}^{\text{FFN}}$ in Eq. (12);
19      Aggregate the FFN experts into $\mathbf{H}^{\text{FFN}}$ via a weighted sum in Eq. (10);
20      $\mathbf{H}^{(l)} \leftarrow Norm(\mathbf{H}^{\text{att}} + \mathbf{O}^{(l)})$;
21      $\mathcal{L}_{\text{rank}} \leftarrow \frac{1}{2}(\mathcal{L}_{\text{rank}}^{\text{att}} + \mathcal{L}_{\text{rank}}^{\text{FFN}})$;
22      $\mathcal{L} \leftarrow \mathcal{L}_{\text{task}} + \lambda \mathcal{L}_{\text{rank}}$;
23      Update $\Theta$ via backpropagation on $\mathcal{L}$;
24 **return** $\mathbf{H}^{(L)}$

---

## C. Table of Symbols

*Table 4.* Notations and their descriptions.

| Notations | Descriptions |
|---|---|
| $\mathcal{G}$ | Continuous-time dynamic graph |
| $I$ | Number of interaction events in the dynamic graph |
| $Seq_v^t$ | Ordered node-centric historical interaction sequence of node $v$ before time $t$ |
| $L$ | Length of the sequence $Seq_v^t$ |
| $d$ | Embedding dimension |
| $\mathbf{X}_N, \mathbf{X}_E, \mathbf{X}_T, \mathbf{X}_I, \mathbf{X}_G$ | Node, edge, temporal, interaction, and geometric features |
| $\mathbf{X}$ | Input embedding of the sequence $Seq_v^t$ |
| $\kappa$ | Curvature of the manifold |
| $\hat{\kappa}_v, \kappa_v^*$ | Routed/Empirical curvature of node $v$ |
| $\mathcal{M}_\kappa$ | stereographic model with constant curvature $\kappa$ |
| $\mathbb{H}, \mathbb{E}, \mathbb{S}$ | Hyperbolic, Euclidean, and spherical manifolds |
| $M$ | Number of curvature-aware experts |
| $g$ | Gating weights of GCRouter |
| $k$ | Number of consecutive selected experts in GCRouter |
| $\mathbf{Q}, \mathbf{K}, \mathbf{V}$ | Query, key, and value representations after Riemannian linear transformation |
| $\mathbf{H}_\kappa^{\text{att}}$ | Riemannian attention representation under curvature $\kappa$ |
| $\mathbf{H}^{\text{att}}$ | Aggregated output of all Riemannian attention experts |
| $\mathbf{O}$ | Output of attention module after residual connection and layer normalization |
| $\mathbf{H}_\kappa^{\text{FFN}}$ | Riemannian FFN representation under curvature $\kappa$ |
| $\mathbf{H}^{\text{FFN}}$ | Aggregated output of all Riemannian FFN experts |
| $\mathbf{s}$ | Sequence-level routing score over all experts |
| $\hat{\mathcal{S}}$ | Routing-score subset with the maximum aggregated routing score |
| $\hat{\mathcal{B}}$ | Selected expert batch corresponding to $\hat{\mathcal{S}}$ |
| $r$ | Euclidean radial coordinate |
| $\mathbf{u}$ | Angular direction on the unit sphere |
| $\rho$ | Geodesic radial coordinate |
| $\theta_{ij}$ | Included angle between $\mathbf{Q}_i$ and $\mathbf{K}_j$ in the polar-coordinate view |
| $\tau$ | Margin hyperparameter in the ranking loss |
| $\lambda$ | Trade-off coefficient balancing the downstream task loss and ranking loss |
| $\oplus_\kappa, \otimes_\kappa$ | Möbius addition, Möbius scalar multiplication under curvature $\kappa$ |
| $\sin_\kappa(\cdot), \cos_\kappa(\cdot)$ | Generalized trigonometric functions associated with curvature $\kappa$ |
| $\tan_\kappa(\cdot), \tan_\kappa^{-1}(\cdot)$ | Generalized tangent function and its inverse under curvature $\kappa$ |
| $\text{Linear}^\kappa(\cdot)$ | Riemannian linear transformations under curvature $\kappa$ |
| $\exp_0^\kappa(\cdot), \log_0^\kappa(\cdot)$ | Exponential map and logarithmic map at the origin on the manifold with curvature $\kappa$ |

## D. Background

### D.1. Riemannian Geometry

A Riemannian manifold $\mathcal{M}$ is a smooth manifold studied in differential geometry. The Riemannian manifold metric $g^\mathcal{M} : \mathcal{T}_\mathbf{x}\mathcal{M}_\kappa \times \mathcal{T}_\mathbf{x}\mathcal{M}_\kappa \to \mathbb{R}$ provides a measure of the distance between points on the manifold. Here, $\mathcal{T}_\mathbf{x}\mathcal{M}$ represents the tangent space of $\mathcal{M}$ at a point $\mathbf{x} \in \mathcal{M}$, and the curvature $\kappa$ quantifies the extent to which the manifold deviates from Euclidean space. In Euclidean space, the curvature $\kappa = 0$ at any point; in hyperbolic space, $\kappa < 0$; and in spherical space, $\kappa > 0$. The $\kappa$-stereographic model $\mathcal{M}_\kappa = \left\{ \mathbf{x} \in \mathbb{R}^d \mid -\kappa \|\mathbf{x}\|^2 < 1 \right\}$ provides a unified representation of constant-curvature manifolds. For Riemannian manifolds with curvature $\kappa$, the transformation of a vector between the manifold and its tangent space is achieved through the logarithmic mapping $\log_\mathbf{x}^\kappa : \mathcal{M}_\kappa \to \mathcal{T}_\mathbf{x}\mathcal{M}_\kappa$, and vice versa, via the exponential mapping

$\exp_{\mathbf{x}}^{\kappa} : \mathcal{T}_{\mathbf{x}}\mathcal{M}_{\kappa} \to \mathcal{M}_{\kappa}$. The logarithmic and exponential mappings of $\mathbf{y}$ at $\mathbf{x}$ is as follows:

$$\log_{\mathbf{x}}^{\kappa}(\mathbf{y}) = \frac{2}{\lambda_{\mathbf{x}}^{\kappa}} \tan_{\kappa}^{-1}(\|(-\mathbf{x}) \oplus_{\kappa} \mathbf{y}\|) \frac{(-\mathbf{x}) \oplus_{\kappa} \mathbf{y}}{\|(-\mathbf{x}) \oplus_{\kappa} \mathbf{y}\|}, \quad \exp_{\mathbf{x}}^{\kappa}(\mathbf{y}) = \mathbf{x} \oplus_{\kappa} \left(\tan_{\kappa}\left(\frac{\|\mathbf{y}\|_{\mathbf{x}}}{2}\right) \frac{\mathbf{y}}{\|\mathbf{y}\|}\right). \tag{17}$$

Here, $\lambda_{\mathbf{x}}^{\kappa} = 2/(1 + \kappa \|\mathbf{x}\|^2)$ is the conformal factor. The $\kappa$-stereographic model is equipped with a conformal Riemannian metric $g_{\mathbf{x}}^{\kappa} = (\lambda_{\mathbf{x}}^{\kappa})^2 \mathbf{I}$, *i.e.*, the inner product at each point is a scalar multiple of the Euclidean one, which preserves angles. When $\mathbf{Mx} \neq \mathbf{0}$, the Riemannian linear transformation is defined as follows:

$$\text{Linear}^{\kappa}(\mathbf{x}) = \tan_{\kappa}\left(\frac{\|\mathbf{Mx}\|}{\|\mathbf{x}\|} \tan_{\kappa}^{-1}(\|\mathbf{x}\|)\right) \frac{\mathbf{Mx}}{\|\mathbf{Mx}\|}, \tag{18}$$

where $\mathbf{M} \in \mathbb{R}^{d' \times d}$ is the weight matrix and $d'$ is the output dimension.

### D.2. Feature Encoding

We adopt the node, edge, and temporal encoding schemes introduced in previous works (Xu et al., 2020; Yu et al., 2023) to obtain node feature $\mathbf{X}_N \in \mathbb{R}^{L \times d_N}$, edge feature $\mathbf{X}_E \in \mathbb{R}^{L \times d_E}$, and temporal feature $\mathbf{X}_T \in \mathbb{R}^{L \times d_T}$, respectively. Specifically, the time interval $\Delta t' = t - t'$ is mapped into temporal features $\mathbf{X}_T = [\Phi(\Delta t_1), ..., \Phi(\Delta t_L)]$ by applying cosine functions, where $\Phi(\Delta t) = [\cos(w_1 \Delta t), ..., \cos(w_{d_T} \Delta t)]$. In addition, we use interaction encoding (Lu et al., 2024) to extract the pairwise information that is crucial for downstream tasks such as link prediction. We store historical pairwise information using temporal walk matrices with time-decayed random feature propagation. For node pairs $(u, v)$, the matrices are updated when a new interaction $(u, v, t)$ occurs:

$$\mathbf{X}_{I,u}^{(l)}(t^+) = \mathbf{X}_{I,u}^{(l)}(t) + e^{\lambda t}\mathbf{X}_{I,v}^{(l-1)}(t), \quad \mathbf{X}_{I,v}^{(l)}(t^+) = \mathbf{X}_{I,v}^{(l)}(t) + e^{\lambda t}\mathbf{X}_{I,u}^{(l-1)}(t), \tag{19}$$

where $t^+$ denotes the time instant immediately after $t$, and $l$ is the temporal-walk length.

### D.3. Graph Properties

(1) **Gromov's $\delta$-hyperbolicity**. This property (Adcock et al., 2013) quantifies the tree-like characteristics of a graph, with lower values of $\delta$ indicating stronger hyperbolicity. In particular, $\delta = 0$ corresponds to a tree structure, and generally low $\delta$ means the graph is close to a tree. In this study, we compute the hyperbolicity of each 3-hop subgraph $G_v$ for each node $v$ at every timestamp. The following outlines the calculation procedure in detail.

To begin, we randomly select four nodes, labeled $a$, $b$, $c$, and $d$, from the subgraph $G_i$. The following quantities are then defined:

$$S_1 = dist(a, b) + dist(c, d), \quad S_2 = dist(a, c) + dist(b, d), \quad S_3 = dist(a, d) + dist(b, c), \tag{20}$$

where $dist$ represents the shortest path length between two nodes. Let $M_1$ and $M_2$ denote the two largest values among $S_1$, $S_2$, and $S_3$. We define the hyperbolicity measure as: $hyp(a, b, c, d) = M_1 - M_2$. The hyperbolicity $\delta$ of the graph $G_v$ is then calculated as half of the maximum value of $hyp$ over all possible 4-tuples $(a, b, c, d)$:

$$\mathcal{C}_{\text{hyperbolicity}}(v) = \delta(G_v) = \max_{a,b,c,d} \frac{hyp(a, b, c, d)}{2}. \tag{21}$$

For subgraphs containing fewer than four nodes, we assign their $\delta(G_v)$ value as 0. Since the naive computation of hyperbolicity requires enumerating all 4-tuples of nodes in the subgraph $G_v$, its time complexity is $O(|\mathcal{V}_{G_v}|^4)$, where $\mathcal{V}_{G_v}$ denotes the node set of $G_v$. To improve efficiency, we adopt the pruning technique from BCCM (Borassi et al., 2015). BCCM is implemented in the Open-Source Mathematical Software System, SageMath, using Python and Cython. We reimplement the BCCM algorithm in pure Python due to compatibility issues with SageMath.

(2) **Neighborhood Growth Rate**. This property describes the rate at which a node neighborhood grows with increasing hop distance, and is used to characterize the local diffusivity, and potential hierarchy of the graph.

$$\mathcal{C}_{\text{growth}}(v) = r_{1 \to 2} + r_{2 \to 3}, \quad r_{1 \to 2} = \frac{|\mathcal{N}_2(v)|}{|\mathcal{N}_1(v)|}, \quad r_{2 \to 3} = \frac{|\mathcal{N}_3(v)|}{|\mathcal{N}_2(v)|}, \tag{22}$$

where $|\mathcal{N}_1(v)|$, $|\mathcal{N}_2(v)|$, and $|\mathcal{N}_3(v)|$ denote the numbers of nodes reachable within one, two, and three hops, respectively.

**(3) Degree Centrality**. This property describes the number of direct connections, representing the local connectivity strength of a node. In social networks, nodes with high degree typically correspond to users with more friends or followers and followee relationships. In communication networks, they may represent routers or devices with more direct links. A larger degree means a node is connected to more immediate neighbors, and is therefore often associated with greater local influence.

$$\mathcal{C}_{\text{degree}}(v) = \deg(v) = |\mathcal{N}_1(v)|, \tag{23}$$

where $\mathcal{N}_1(v)$ denotes the set of 1-hop neighbors of node $v$.

**(4) Betweenness Centrality**. This property is the sum of the fraction of all-pairs shortest paths that pass through $v$. It measures the extent to which a node serves as an intermediary that connects other nodes. Nodes with high betweenness centrality often link different communities or subgraphs as critical bridges, such as transportation hubs, critical routers, or connectors between social groups.

$$\mathcal{C}_{\text{betweenness}}(v) = \sum_{s,t \in \mathcal{V}_{G_v}} \frac{\pi(s,t \mid v)}{\pi(s,t)}, \tag{24}$$

where $\mathcal{V}_{G_v}$ is the node set of the subgraph $G_v$, $\pi(s,t)$ denotes the number of shortest paths between $s$ and $t$, and $\pi(s,t \mid v)$ denotes the number of such paths that pass through $v$.

**(5) Closeness Centrality**. This property captures how centrally a node is positioned in the network, quantified by the average shortest-path distance from the node to all others. Nodes with high closeness centrality can reach most other nodes in the network within a small number of hops, and thus facilitate efficient global information diffusion.

$$\mathcal{C}_{\text{closeness}}(v) = \frac{n-1}{\sum_{u=1}^{n-1} dist(v,u)}, \tag{25}$$

where $dist(v,u)$ is the shortest-path distance between nodes $v$ and $u$, and $n-1$ is the number of nodes reachable from $v$.

**(6) Clustering Coefficient**. This property describes the fraction of possible triangles incident to a node that exist, reflecting the local clustering tendency within the neighborhood. A high clustering coefficient indicates that the neighbors are more likely to form tightly connected triangular structures. In contrast, a low value implies sparser connections, which is characteristic of tree-like or more randomly organized topologies.

$$\mathcal{C}_{\text{clustering}}(v) = \frac{2T(v)}{\deg(v)\big(\deg(v) - 1\big)}, \tag{26}$$

where $T(v)$ is the number of triangles through node $v$.

**(7) Square Clustering Coefficient**. This property characterizes the tendency of a node neighborhood to form four-node cycles, capturing higher-order local structural correlations beyond triangle-based connectivity.

$$\mathcal{C}_{\text{square}}(v) = \frac{\sum_{u=1}^{\deg(v)} \sum_{w=u+1}^{\deg(v)} q_v(u,w)}{\sum_{u=1}^{\deg(v)} \sum_{w=u+1}^{\deg(v)} \big[a_v(u,w) + q_v(u,w)\big]}, \tag{27}$$

where $q_v(u,w)$ counts the common neighbors shared by $u$, $w$, and $v$ (*i.e.*, squares), and $a_v(u,w) = (\deg(u) - (1 + q_v(u,w) + \theta_{uw})) + (\deg(w) - (1 + q_v(u,w) + \theta_{uw}))$, with $\theta_{uw}$ equals 1 if $u$ and $w$ are connected and 0 otherwise.

## E. Fast Riemannian Attention

### E.1. Riemannian Polar Coordinate

For a vector $\mathbf{x} \in \mathcal{M}_\kappa$, the Riemannian metric $g_{\mathbf{x}}^\kappa : \mathcal{T}_{\mathbf{x}}\mathcal{M}_\kappa \times \mathcal{T}_{\mathbf{x}}\mathcal{M}_\kappa \to \mathbb{R}$ defines inner products on tangent spaces. Specifically, for $\mathbf{v}_1, \mathbf{v}_2 \in \mathcal{T}_{\mathbf{x}}\mathcal{M}_\kappa$, the Riemannian metric is conformal to the Euclidean metric $g_{\mathbf{x}}^\kappa(\mathbf{v}_1, \mathbf{v}_2) = \left(\lambda_{\mathbf{x}}^\kappa\right)^2 \langle \mathbf{v}_1, \mathbf{v}_2 \rangle_{\mathbb{E}}$, where $\lambda_{\mathbf{x}}^\kappa = 2/(1 + \kappa \|\mathbf{x}\|^2)$ is the conformal factor and $\langle \cdot, \cdot \rangle_{\mathbb{E}}$ is the Euclidean inner product. This implies that the Riemannian metric is conformally flat, differing from the Euclidean inner product at each point by a positive scalar factor $\lambda_{\mathbf{x}}^\kappa$, while preserving the angles of the Euclidean space. The length of a vector $\mathbf{v} \in \mathcal{T}_{\mathbf{x}}\mathcal{M}_\kappa$ induced by the Riemannian metric is given by $\|\mathbf{v}\|_{\mathbf{x},g} = \sqrt{g_{\mathbf{x}}^\kappa(\mathbf{v}, \mathbf{v})}$.

Consider a nonzero vector $\mathbf{x} \in \mathbb{R}^d \setminus \{\mathbf{0}\}$ on the manifold $\mathcal{M}_\kappa = \{\mathbf{x} \mid -\kappa\|\mathbf{x}\|^2 < 1\}$ (Bachmann et al., 2020). Adopting Euclidean coordinates on the manifold, we express the polar coordinates as:

$$\mathbf{x} = r\,\mathbf{u}, \quad r = \|\mathbf{x}\|, \quad \mathbf{u} = \frac{\mathbf{x}}{\|\mathbf{x}\|}. \tag{28}$$

Here, $r \in \mathbb{R}_+$ represents the Euclidean radial coordinate, and $\mathbf{u}$ specifies the angular direction on the unit sphere $\mathbb{S}^{d-1}$. On the other hand, the Riemannian distance from the origin $\mathbf{0}$ to $\mathbf{x} \in \mathcal{M}_\kappa$, also referred to as geodesic radial coordinate, is denoted by $\rho \in \mathbb{R}_+$. Since the Riemannian metric is conformal to the Euclidean metric, the geodesic from the origin $\mathbf{0}$ to $\mathbf{x}$ can be parameterized as $\gamma(t) := \mathbf{x}(t) = t\mathbf{u}$, where $t \in [0, r]$. Its tangent vector is given by $\dot{\gamma}(t) = \frac{d\mathbf{x}(t)}{dt} = \frac{d}{dt}(t\mathbf{u}) = \mathbf{u}$. Substituting $\mathbf{v} = \dot{\gamma}(t) = \mathbf{u}$ into the length defined by the Riemannian metric, we obtain $\|\dot{\gamma}(t)\|_{\gamma(t),g} = \sqrt{g^\kappa_{\gamma(t)}(u,u)} = \sqrt{\left(\frac{2}{1+\kappa t^2}\right)^2} = \frac{2}{1+\kappa t^2}$. The Riemannian distance $\rho$ represents the length of the geodesic:

$$\rho = \int_0^r \|\dot{\gamma}(t)\|_{\gamma(t),g}\, dt = \int_0^r \frac{2}{1 + \kappa t^2}\, dt. \tag{29}$$

When $\kappa > 0$, we have $\rho = 2 \cdot \frac{1}{\sqrt{\kappa}} \int_0^{\sqrt{\kappa}r} \frac{1}{1+(\sqrt{\kappa}t)^2}\sqrt{\kappa}dt = \frac{2}{\sqrt{\kappa}}\arctan(\sqrt{\kappa}r)$. When $\kappa = 0$, the expression simplifies to $\rho = \int_0^r \frac{2}{1+0}dt = 2\int_0^r dt = 2r$. In addition, when $\kappa < 0$, we obtain $\rho = 2\int_0^r \frac{1}{1-(\sqrt{-\kappa}t)^2}dt = \frac{2}{\sqrt{-\kappa}}\tanh^{-1}(\sqrt{-\kappa}r)$. According to the definition of the generalized tangent function $\tan_\kappa(\cdot)$ in the Preliminary section, $\rho$ is unified as $\rho = 2\tan_\kappa^{-1}(r)$.

### E.2. Riemannian Law of Cosines

**Hyperbolic.** Consider a hyperbolic triangle $ABO$ with side lengths $OA = a$, $OB = b$, and $AB = c$, as illustrated in Figure 3(a). While the figure is depicted in the two-dimensional Poincaré ball model ($d = 2$), the following hyperbolic law of cosines holds in arbitrary dimensions ($d \geq 2$):

$$\cosh(\sqrt{-\kappa}\,c) = \cosh(\sqrt{-\kappa}\,a)\cosh(\sqrt{-\kappa}\,b) - \sinh(\sqrt{-\kappa}\,a)\sinh(\sqrt{-\kappa}\,b)\cos\theta_{ij}. \tag{30}$$

Note that the hyperbolic law of cosines does not require the point $O$ to be at the origin. We place $O$ at the origin only for the derivation of Eq. (15). In Eq. (15), the geodesic radial coordinate $\rho_i^{\mathbf{Q}}$ of $\mathbf{Q}_i$ corresponds to $a$, and the geodesic radial coordinate $\rho_j^{\mathbf{K}}$ of $\mathbf{K}_j$ corresponds to $b$. The angle between them is $\theta_{ij}$, and the desired distance $\mathcal{D}_\kappa(\mathbf{Q}_i, \mathbf{K}_j)$ corresponds to $c$. Consequently, the geodesic distance can be written as:

$$\mathcal{D}_\kappa(\mathbf{Q}_i, \mathbf{K}_j) = \frac{1}{\sqrt{-\kappa}}\operatorname{arcosh}(\cosh(\sqrt{-\kappa}\,c)). \tag{31}$$

**Spherical.** The spherical triangle $ABO$ is shown in Figure 3(b). The spherical law of cosines is also valid in arbitrary dimensions ($d \geq 2$):

$$\cos(\sqrt{\kappa}\,c) = \cos(\sqrt{\kappa}\,a)\cos(\sqrt{\kappa}\,b) + \sin(\sqrt{\kappa}\,a)\sin(\sqrt{\kappa}\,b)\cos\theta_{ij}, \tag{32}$$

where the geodesic radius $\rho_i^{\mathbf{Q}}$ of $\mathbf{Q}_i$ corresponds to $a$, and the geodesic radius $\rho_j^{\mathbf{K}}$ of $\mathbf{K}_j$ corresponds to $b$. The angle between them is $\theta_{ij}$, and the desired distance $\mathcal{D}_\kappa(\mathbf{Q}_i, \mathbf{K}_j)$ corresponds to $c$. The geodesic distance can be written as:

$$\mathcal{D}_\kappa(\mathbf{Q}_i, \mathbf{K}_j) = \frac{1}{\sqrt{\kappa}}\arccos(\cos(\sqrt{\kappa}\,c)). \tag{33}$$

For a stereographic projection model with curvature $\kappa$, the radius is $R = 1/\sqrt{\kappa}$, so the distance $c = \frac{2}{\sqrt{\kappa}}\arctan(\sqrt{\kappa}\,R)$ lies in $[0, \pi/\sqrt{\kappa}]$. Since $\arccos : [-1, 1] \to [0, \pi]$ is strictly monotone and bijective on this interval, the expression in Eq. (33) does not introduce any multivalued ambiguity.

**Euclidean.** As $\kappa \to 0+$ and $\kappa \to 0-$, the geodesic distance $\mathcal{D}_\kappa(\mathbf{Q}_i, \mathbf{K}_j)$ converges to the Euclidean expression $2\|\mathbf{Q}_i - \mathbf{K}_j\|$.

(1) $\kappa > 0$. We apply the Taylor expansions for the sine and cosine functions:

$$\sin(\sqrt{\kappa}x) = \sqrt{\kappa}x + O(\kappa^{3/2}), \quad \cos(\sqrt{\kappa}x) = 1 - \tfrac{1}{2}\kappa x^2 + O(\kappa^2). \tag{34}$$

By substituting these expansions into Eq. (15), we obtain the following expression for the geodesic distance:

$$1 - \tfrac{1}{2}\kappa \, \mathcal{D}_\kappa^2(\mathbf{Q}_i, \mathbf{K}_j) + O(\kappa^2) = 1 - \tfrac{1}{2}\kappa\left((\rho_i^{\mathbf{Q}})^2 + (\rho_j^{\mathbf{K}})^2\right) + \mathrm{sign}_\kappa \, \kappa \, \rho_i^{\mathbf{Q}} \rho_j^{\mathbf{K}} \cos\theta_{ij} + O(\kappa^2). \tag{35}$$

As $\kappa \to 0^+$, this expression simplifies to the Euclidean form:

$$\mathcal{D}_0^2(\mathbf{Q}_i, \mathbf{K}_j) = (\rho_i^{\mathbf{Q}})^2 + (\rho_j^{\mathbf{K}})^2 - 2\rho_i^{\mathbf{Q}} \rho_j^{\mathbf{K}} \cos\theta_{ij}. \tag{36}$$

(2) $\kappa < 0$. We use the Taylor expansions for the hyperbolic sine and cosine functions:

$$\sinh(\sqrt{-\kappa}\,x) = \sqrt{-\kappa}\,x + O(-\kappa^{3/2}), \quad \cosh(\sqrt{-\kappa}\,x) = 1 - \tfrac{1}{2}\kappa x^2 + O(\kappa^2). \tag{37}$$

By substituting these expansions into Eq. (15), we obtain the following expression for the geodesic distance:

$$1 - \tfrac{1}{2}\kappa \, \mathcal{D}_\kappa^2(\mathbf{Q}_i, \mathbf{K}_j) + O(\kappa^2) = 1 - \tfrac{1}{2}\kappa\left((\rho_i^{\mathbf{Q}})^2 + (\rho_j^{\mathbf{K}})^2\right) - \mathrm{sign}_\kappa \, \kappa \, \rho_i^{\mathbf{Q}} \rho_j^{\mathbf{K}} \cos\theta_{ij} + O(\kappa^2). \tag{38}$$

As $\kappa \to 0^-$, this expression simplifies to the Euclidean form:

$$\mathcal{D}_0^2(\mathbf{Q}_i, \mathbf{K}_j) = (\rho_i^{\mathbf{Q}})^2 + (\rho_j^{\mathbf{K}})^2 - 2\rho_i^{\mathbf{Q}} \rho_j^{\mathbf{K}} \cos\theta_{ij}. \tag{39}$$

Recall that the radial coordinate is $r = \|\mathbf{x}\|$, and the corresponding geodesic radial coordinate is $\rho = 2\tan_\kappa^{-1}(r)$. The generalized tangent function $\tan_\kappa(\cdot)$ is defined as $\frac{1}{\sqrt{\kappa}}\tan(\sqrt{\kappa}\,r)$ if $\kappa > 0$, $\frac{1}{\sqrt{-\kappa}}\tanh(\sqrt{-\kappa}\,r)$ if $\kappa < 0$, and $r$ if $\kappa = 0$. Using the approximations $\tan(y) = y + O(y^3)$ and $\tanh(y) = y + O(y^3)$ as $y \to 0$, we have: $\tan_\kappa(r) \to r$ as $\kappa \to 0$. Thus, both $\rho_i^{\mathbf{Q}}$ and $\rho_j^{\mathbf{K}}$ converge to twice the Euclidean distances $2\|\mathbf{Q}_i\|$ and $2\|\mathbf{K}_i\|$, respectively. Finally, the angular term is given by the cosine similarity between $\mathbf{Q}_i$ and $\mathbf{K}_j$: $\cos\theta_{ij} = \frac{\langle \mathbf{Q}_i, \mathbf{K}_j \rangle}{\|\mathbf{Q}_i\| \, \|\mathbf{K}_j\|}$. Substituting these limits into Eq. (36), we obtain:

$$\begin{aligned} \mathcal{D}_0^2(\mathbf{Q}_i, \mathbf{K}_j) &= (2\|\mathbf{Q}_i\|)^2 + (2\|\mathbf{K}_j\|)^2 - 2(2\|\mathbf{Q}_i\|)(2\|\mathbf{K}_j\|)\frac{\langle \mathbf{Q}_i, \mathbf{K}_j \rangle}{\|\mathbf{Q}_i\| \, \|\mathbf{K}_j\|} \\ &= 4\|\mathbf{Q}_i\|^2 + 4\|\mathbf{K}_j\|^2 - 8\langle \mathbf{Q}_i, \mathbf{K}_j \rangle \\ &= 4\|\mathbf{Q}_i - \mathbf{K}_j\|^2. \end{aligned} \tag{40}$$

## F. Computational Cost

In this section, we provide a detailed analysis of the time complexity and floating-point operation counts (FLOPs) for Euclidean, Riemannian, and our fast Riemannian attention in dynamic graphs. Our analysis indicates that, although Riemannian and Euclidean attention share the same time complexity, Riemannian attention incurs significantly higher FLOPs due to the computational cost of operations on Riemannian manifolds. In contrast, our fast Riemannian attention achieves a significant reduction in FLOPs. Furthermore, we analyze the existing Riemannian acceleration techniques in hyperbolic GNNs and highlight their limitations. In the following, we provide discussions on: (1) time complexity and FLOPs, and (2) current Riemannian acceleration methods and their limitations.

### F.1. Time complexity and FLOPs

The input to the attention module of the dynamic graph is $\mathbf{X} \in \mathbb{R}^{L \times d}$, where $L$ represents the length of the historical interaction sequence $Seq_v^t$, and $d$ denotes the embedding dimension. Next, we analyze the time complexity and FLOPs of the attention modules in the three different variants.

#### Euclidean Attention

*Step1*: Linear Transformation. Linear transformations defined in the Euclidean space are applied to the input $\mathbf{X}$ to obtain the query $\mathbf{Q} = \mathrm{Linear}_q(\mathbf{X}) \in \mathbb{R}^{L \times d}$, key $\mathbf{K} = \mathrm{Linear}_k(\mathbf{X}) \in \mathbb{R}^{L \times d}$, and value $\mathbf{V} = \mathrm{Linear}_v(\mathbf{X}) \in \mathbb{R}^{L \times d}$.

*Step2*: Similarity Score Computation. Euclidean attention uses the dot-product $\alpha_{ij} = \mathbf{Q}_i \mathbf{K}_j^\top$ to compute the similarity score.

*Step3*: Similarity Score Normalization. Softmax function $w_{ij} = \frac{\exp(\alpha_{ij}/\sqrt{d})}{\sum_{l \in Seq_v^t} \exp(\alpha_{il}/\sqrt{d})}$ is used to normalize the similarity score.

*Table 5.* Time complexity and FLOPs of Euclidean attention, Riemannian attention, and our fast Riemannian attention.

| Methods | Time Complexity | | | FLOPs | | |
|---|---|---|---|---|---|---|
| | Euclidean Attention | Riemannian Attention | Fast Riemannian Attention | Euclidean Attention | Riemannian Attention | Fast Riemannian Attention |
| *Step1* | $O(Ld^2)$ | $O(Ld^2)$ | $O(Ld^2)$ | $3Ld(2d-1)$ | $3L(2d^2+7d+13)$ | $3L(2d^2+7d+13)$ |
| *Step2* | $O(L^2d)$ | $O(L^2d)$ | $O(L^2d)$ | $L^2(2d-1)$ | $L^2(18d+15)$ | $L^2(11d+10)$ |
| *Step3* | $O(L^2)$ | $O(L^2)$ | $O(L^2)$ | $4L(L-1)$ | $4L(L-1)$ | $4L(L-1)$ |
| *Step4* | $O(L^2d)$ | $O(L^2d)$ | $O(L^2d)$ | $Ld(2L-1)$ | $L(7Ld+10L-d)$ | $L^2(2d+1)$ |
| Total | $O(L^2d+Ld^2)$ | $O(L^2d+Ld^2)$ | $O(L^2d+Ld^2)$ | $L(4Ld+6d^2+3L-4d-4)$ | $L(25Ld+6d^2+29L+20d+35)$ | $L(13Ld+6d^2+15L+21d+35)$ |

*Step4*: Value Aggregation. The weighted values $\mathbf{V}_j$ are aggregated using the normalized attention weights $w_{ij}$ to compute the output $\mathbf{H}_i^{\text{att}} = \sum_{j \in Seq_v^t} w_{ij} \mathbf{V}_j \in \mathbb{R}^d$. Here, $\mathbf{H}^{\text{att}} \in \mathbb{R}^{L \times d}$ represents the result of applying attention to the input $\mathbf{X}$.

**Conventional Riemannian attention**

*Step1*: Linear Transformation. As shown in Eq. (3), the input $\mathbf{X}$ is first mapped into the Riemannian manifold via the exponential mapping, followed by a Riemannian linear transformation as defined in Eq. (18).

*Step2*: Similarity Score Computation. The similarity measure is based on the negative geodesic distance, as described in Eq. (4). The computation of this distance requires the use of Möbius addition operations as defined in Eq. (5).

*Step3*: Similarity Score Normalization. The normalization of the similarity score follows the same principles as those in Euclidean space.

*Step4*: Value Aggregation. The first step in value aggregation is the application of Möbius scalar multiplication to obtain $\tilde{V}_{ij} = w_{ij} \otimes_\kappa \mathbf{V}_j$, where $\otimes\kappa$ denotes the Möbius scalar multiplication as defined in Eq. (7). Then, the aggregated values are computed using Möbius addition as $\mathbf{H}_i^{\text{att}_\kappa} = \sum_{j \in Seq_v^t}^{\oplus_\kappa} \tilde{V}_{ij} = \left( \left( \left( \tilde{V}_{i1} \oplus_\kappa \tilde{V}_{i2} \right) \oplus_\kappa \tilde{V}_{i3} \right) \oplus_\kappa \cdots \right)$. Since Möbius addition is neither commutative nor associative, the sum $\sum_{j \in Seq_v^t}^{\oplus_\kappa} \tilde{V}_{ij}$ must be computed in sequence. To improve computational efficiency, prior work (Chami et al., 2019; Xu et al., 2024; Yang et al., 2025) projects the embeddings into Euclidean space, aggregates them, and then maps the result back to the Riemannian manifold, as described in Eq. (6). This accelerated approach is referred to as conventional Riemannian Attention. For a fair comparison with our method, we focus on the portion before mapping back to the Riemannian manifold $\mathbf{H}_i^{\text{att}_\kappa} = \sum_{j \in Seq_v^t} \log_{\mathbf{0}}^\kappa (w_{ij} \otimes_\kappa \mathbf{V}_j)$ when comparing computational complexity and FLOPs.

**Fast Riemannian attention**

*Step1*: Linear Transformation. This step is identical to *Step1* of conventional Riemannian attention.

*Step2*: Similarity Score Computation. We begin by performing a polar decomposition on $\mathbf{Q}$, $\mathbf{K}$, and $\mathbf{V}$, as outlined in Eq. (14). Subsequently, as demonstrated in Eq. (15), we apply the Riemannian law of cosines to compute the negative geodesic distance, which serves as the similarity measure.

*Step3*: Similarity Score Normalization. The normalization of the similarity score follows the same principles as those in Euclidean space.

*Step4*: Value Aggregation. We simplify the value aggregation in Eq. (16) via polar decomposition.

**Comparison.** Existing studies on curvature-aware Riemannian graph transformers primarily focus on static graphs. While STGN$^h$ (Xu et al., 2024) addresses dynamic graphs, it only analyzes the computational complexity and does not provide an analysis of FLOPs. Our work bridges this gap. We define a single floating-point addition, multiplication, square root, or any elementary transcendental function (e.g., sin, cos, tanh, atan, acos, acosh, or atanh) as one FLOP. As summarized in Table 5, all three methods exhibit the same time complexity of $O(L^2d + Ld^2)$, but their FLOP counts differ substantially. We examine whether our fast Riemannian attention could ever be slower than the conventional Riemannian attention. Specifically, we analyze the inequality: $L(13Ld + 6d^2 + 15L + 21d + 35) > L(25Ld + 6d^2 + 29L + 20d + 35)$. After simplification, this inequality reduces to $\frac{1}{L} - \frac{14}{d} > 12$. Since $L$, and $d$ are all positive integers, the inequality can never be satisfied. Therefore, our method is always faster than the conventional method under these conditions.

## F.2. Other acceleration methods

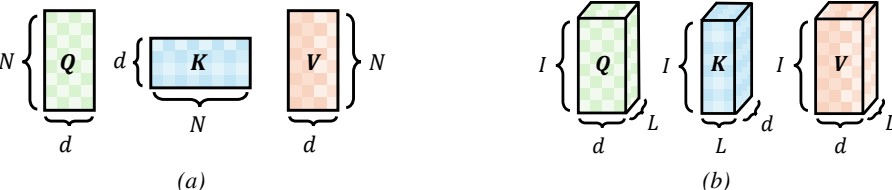

*Figure 7.* Illustration of the query $\mathbf{Q}$, key $\mathbf{K}$, and value $\mathbf{V}$ shapes in (a) static graph attention and (b) dynamic graph attention.

The batch input to static graph attention is $\mathbf{X} \in \mathbb{R}^{N \times d}$, where $N$ denotes the total number of nodes in the graph and $d$ represents the embedding dimension. Inspired by FLatten Transformer (Han et al., 2023), previous Riemannian methods (Cho et al., 2023; Yang et al., 2024) alter the computation sequence from $(\mathbf{Q}^\top \mathbf{K})\mathbf{V}$ to $\mathbf{Q}(\mathbf{K}^\top \mathbf{V})$ to achieve linear attention. This reformulation reduces the time complexity of the attention module from $O(N^2 d)$ to $O(Nd^2)$. For additional implementation details, please refer to the original paper.

In contrast, the batch input to dynamic graph attention is $\mathbf{X} \in \mathbb{R}^{I \times L \times d}$, where $I$ is the total number of interaction events, $L$ represents the length of the ordered sequence consisting of a node and its neighbors, and $d$ denotes the embedding dimension. Due to the different input dimensionality, applying linear attention alters the computational complexity from $O(IL^2 d)$ to $O(ILd^2)$. However, in typical settings, the sequence length $L$ (e.g., 32 or 64) is smaller than the embedding dimension $d$ (e.g., 200). As a result, this strategy does not lead to a reduction in computational complexity. In particular, throughout the main text, the input $\mathbf{X} \in \mathbb{R}^{L \times d}$ refers to a single node-centric sequence associated with one interaction event and should be distinguished from the batched input $\mathbf{X} \in \mathbb{R}^{I \times L \times d}$.

# G. Datasets and Implementation Details

The detailed statistics of eight real-world datasets are summarized in Table 6. In the table, #N&L Feat denotes the dimensionality of the node and link features. For each dataset, we measure the average Gromov's $\delta$-hyperbolicity of nodes at both the initial and final timestamps. We compute hyperbolicity on 3-hop neighborhoods and limit each node to at most 32 historical neighbors.

*Table 6.* Statistics of the datasets.

| Datasets | Domains | #Nodes | #Links | #N&L Feat | Hyperbolicity | Bipartite | Duration | Unique Steps | Time Granularity |
|---|---|---|---|---|---|---|---|---|---|
| Wikipedia | Social | 9,227 | 157,474 | – & 172 | 0.006→0.135 | True | 1 months | 152,757 | Unix timestamps |
| Reddit | Interaction | 10,984 | 672,447 | – & 172 | 0.024→1.521 | True | 1 months | 669,065 | Unix timestamps |
| MOOC | Interaction | 7,144 | 411,749 | – & 4 | 0.019→1.850 | True | 17 months | 345,600 | Unix timestamps |
| LastFM | Interaction | 1,980 | 1,293,103 | – & – | 0.613→1.684 | True | 1 month | 1,283,614 | Unix timestamps |
| Enron | Social | 184 | 125,235 | – & – | 0.315→0.829 | False | 3 years | 22,632 | Unix timestamps |
| US Legis. | Politics | 225 | 60,396 | – & 1 | 0.007→0.391 | False | 12 congresses | 12 | congresses |
| UN Trade | Economics | 255 | 507,497 | – & 1 | 0.092→1.896 | False | 32 years | 32 | years |
| UN Vote | Politics | 201 | 1,035,742 | – & 1 | 0.000→0.627 | False | 72 years | 72 | years |

### G.1. Empirical Evidence of Heterogeneous and Evolving Local Topology

To characterize the variation of local topology, we employ the mean $\mu$, variance $\delta$, and coefficient of variation ($CV = \delta/\mu$) as quantitative measures, where a larger $CV$ indicates stronger relative fluctuation across nodes or over time. To examine cross-node geometric heterogeneity at a given timestamp, we compute these statistics at every $10\%$ interval of the full temporal span, as shown in Figure 8. Compared with Barabási-Albert and Erdős–Rényi graphs of similar size and average degree, our real-world datasets exhibit substantially higher $CV$ values, confirming that real temporal graphs possess greater cross-node heterogeneity. To further investigate temporal evolution, we compute per-node $CV$ across its active timestamps. The observed clear variation (Figure 9) demonstrates that local topology of individual nodes also change significantly over time. The above empirical evidence justifies the design of our MoE.

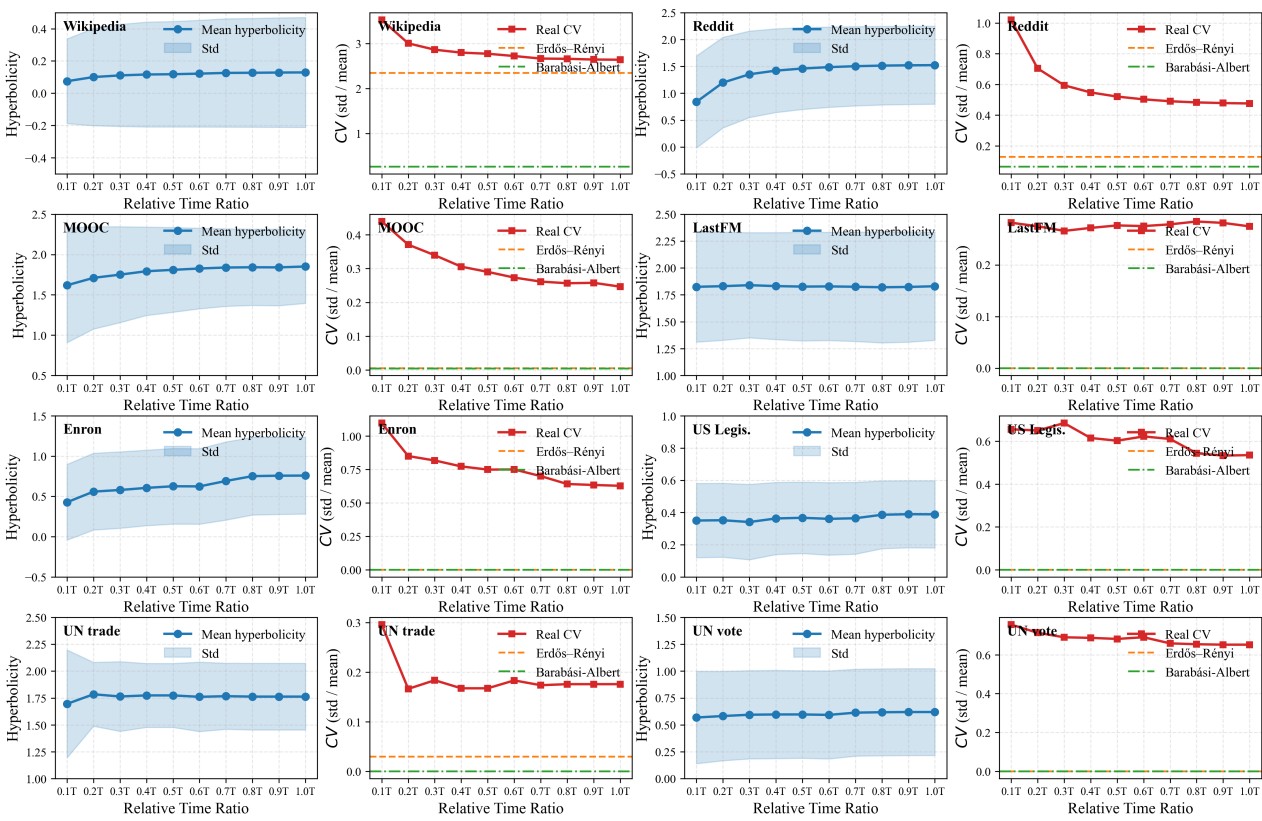

*Figure 8.* Illustration of cross-node topology heterogeneity at the same timestamp. Compared with Barabási-Albert and Erdős–Rényi graphs of similar size and average degree, real-world datasets exhibit substantially larger variation.

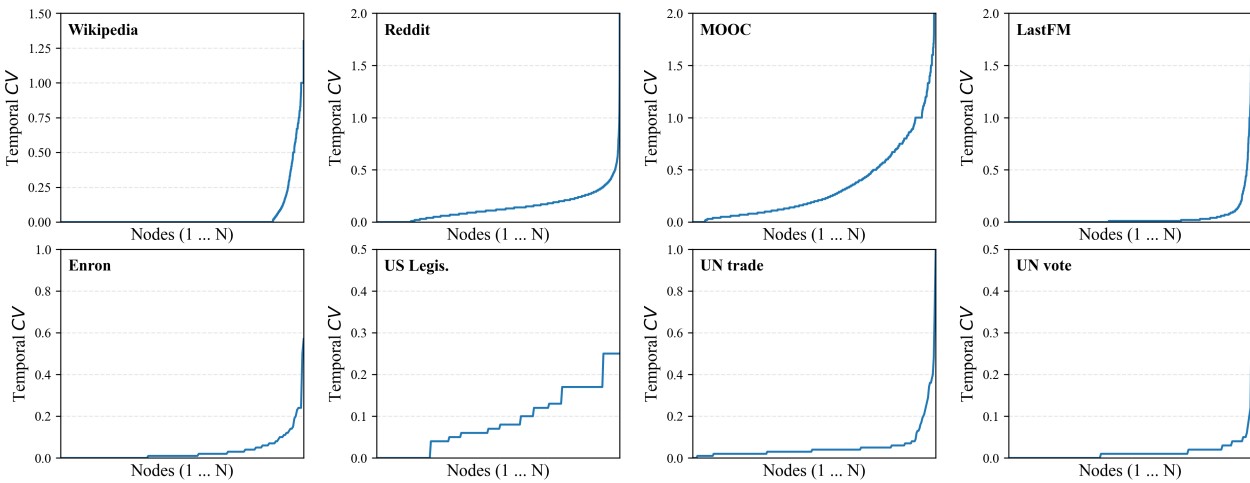

*Figure 9.* Illustration of temporal dynamics in local topology. For each node, we compute its coefficient of variation ($CV = \delta/\mu$) across all timestamps where it is active.

## H. Supplementary Experimental Results

### H.1. Results on Link Prediction

We report AUC-ROC for dynamic link prediction in the transductive setting in Table 7. For the inductive setting, AP and AUC-ROC results are provided in Table 8 and Table 9, respectively. Overall, DyGMoCE remains the top-performing method in terms of average rank, followed consistently by TPNet as the second-best baseline. For the third position, DyGFormer shows stronger overall competitiveness across AUC-ROC metrics, while STGN$^h$ ranks third in inductive AP.

### H.2. Results on Efficiency

Computational efficiency results on the remaining datasets are reported in Table 10. Fast RA substantially improves the efficiency of conventional RA, reducing running time by up to $32.5\%$ and memory usage by up to $53.7\%$ across different datasets and input lengths.

*Table 7.* AUC-ROC for transductive dynamic link prediction with random, historical, and inductive negative sampling strategies (NSS). The best and runner-up results are emphasized by **bold** and underlined fonts. Rank indicates the average rank across datasets.

| NSS | Methods | Wikipedia | Reddit | MOOC | LastFM | Enron | US Legis. | UN Trade | UN Vote | Rank |
|---|---|---|---|---|---|---|---|---|---|---|
| rnd | TGAT | $96.67_{\pm0.07}$ | $98.47_{\pm0.02}$ | $87.11_{\pm0.19}$ | $71.59_{\pm0.18}$ | $68.89_{\pm1.10}$ | $75.84_{\pm1.99}$ | $64.01_{\pm0.12}$ | $52.83_{\pm1.12}$ | 8.00 |
| | GraphMixer | $96.92_{\pm0.03}$ | $97.17_{\pm0.02}$ | $84.01_{\pm0.17}$ | $73.53_{\pm0.12}$ | $84.38_{\pm0.21}$ | $76.96_{\pm0.79}$ | $65.52_{\pm0.51}$ | $52.46_{\pm0.27}$ | 7.75 |
| | DyGFormer | $98.91_{\pm0.02}$ | $99.15_{\pm0.01}$ | $87.91_{\pm0.58}$ | $93.05_{\pm0.10}$ | $93.33_{\pm0.13}$ | $77.90_{\pm0.58}$ | $70.20_{\pm1.44}$ | $57.12_{\pm0.62}$ | 4.50 |
| | RepeatMixer | $98.86_{\pm0.03}$ | $98.96_{\pm0.01}$ | $85.36_{\pm0.11}$ | $91.50_{\pm0.02}$ | $92.85_{\pm0.27}$ | $75.42_{\pm0.37}$ | $69.54_{\pm0.26}$ | $56.62_{\pm0.04}$ | 6.38 |
| | TPNet | $99.32_{\pm0.02}$ | $99.22_{\pm0.01}$ | $\underline{95.62_{\pm0.10}}$ | $\underline{94.40_{\pm0.10}}$ | $94.35_{\pm0.15}$ | $\underline{86.20_{\pm0.09}}$ | $\underline{89.30_{\pm0.17}}$ | $\underline{79.75_{\pm0.05}}$ | 2.13 |
| | STGN$^h$ | $98.08_{\pm0.03}$ | $98.94_{\pm0.02}$ | $87.42_{\pm0.12}$ | $87.46_{\pm0.12}$ | $89.97_{\pm0.62}$ | $76.72_{\pm0.84}$ | $70.49_{\pm0.26}$ | $63.74_{\pm0.75}$ | 5.63 |
| | TIDFormer | $99.25_{\pm0.02}$ | $99.12_{\pm0.01}$ | $92.85_{\pm0.13}$ | $93.15_{\pm0.09}$ | $92.87_{\pm0.19}$ | $68.11_{\pm0.14}$ | $51.32_{\pm2.96}$ | $59.61_{\pm1.15}$ | 5.38 |
| | TAMI | $99.24_{\pm0.01}$ | $\underline{99.24_{\pm0.01}}$ | $87.99_{\pm0.37}$ | $93.44_{\pm0.11}$ | $93.70_{\pm0.31}$ | $74.65_{\pm0.51}$ | $69.46_{\pm0.21}$ | $60.99_{\pm2.18}$ | 4.25 |
| | DyGMoCE | $\mathbf{99.40_{\pm0.02}}$ | $\mathbf{99.29_{\pm0.02}}$ | $\mathbf{96.08_{\pm0.23}}$ | $\mathbf{95.62_{\pm0.11}}$ | $\mathbf{95.21_{\pm0.16}}$ | $\mathbf{89.92_{\pm0.43}}$ | $\mathbf{90.81_{\pm0.15}}$ | $\mathbf{82.44_{\pm0.03}}$ | 1.00 |
| hist | TGAT | $82.87_{\pm0.22}$ | $79.33_{\pm0.16}$ | $80.81_{\pm0.67}$ | $64.27_{\pm0.26}$ | $61.85_{\pm1.43}$ | $73.47_{\pm5.25}$ | $60.37_{\pm0.68}$ | $53.95_{\pm3.15}$ | 7.63 |
| | GraphMixer | $\mathbf{87.68_{\pm0.17}}$ | $77.80_{\pm0.12}$ | $76.68_{\pm1.40}$ | $64.21_{\pm0.73}$ | $75.27_{\pm1.14}$ | $85.17_{\pm0.70}$ | $63.20_{\pm1.54}$ | $52.61_{\pm1.44}$ | 7.00 |
| | DyGFormer | $78.80_{\pm1.95}$ | $80.54_{\pm0.29}$ | $87.04_{\pm0.35}$ | $78.78_{\pm0.35}$ | $76.55_{\pm0.52}$ | $90.77_{\pm1.96}$ | $73.86_{\pm1.13}$ | $64.27_{\pm1.78}$ | 5.25 |
| | RepeatMixer | $86.30_{\pm0.67}$ | $\underline{83.50_{\pm0.40}}$ | $87.19_{\pm5.92}$ | $79.98_{\pm4.65}$ | $67.39_{\pm16.4}$ | $88.24_{\pm0.17}$ | $65.35_{\pm18.6}$ | $58.94_{\pm1.11}$ | 4.63 |
| | TPNet | $81.07_{\pm0.36}$ | $81.61_{\pm0.15}$ | $\underline{90.94_{\pm1.08}}$ | $\underline{84.92_{\pm0.87}}$ | $79.60_{\pm0.89}$ | $\underline{95.88_{\pm0.60}}$ | $\underline{89.08_{\pm0.54}}$ | $\underline{77.49_{\pm0.19}}$ | 2.88 |
| | STGN$^h$ | $83.24_{\pm0.21}$ | $82.34_{\pm0.10}$ | $83.16_{\pm0.43}$ | $80.04_{\pm0.24}$ | $76.12_{\pm0.98}$ | $86.40_{\pm2.56}$ | $67.91_{\pm0.33}$ | $69.47_{\pm2.04}$ | 4.75 |
| | TIDFormer | $81.79_{\pm1.22}$ | $80.74_{\pm0.35}$ | $87.90_{\pm1.19}$ | $82.34_{\pm1.50}$ | $78.02_{\pm1.65}$ | $66.29_{\pm5.92}$ | $49.29_{\pm1.58}$ | $43.41_{\pm2.01}$ | 6.00 |
| | TAMI | $78.08_{\pm3.96}$ | $80.94_{\pm0.18}$ | $83.62_{\pm1.69}$ | $73.31_{\pm0.73}$ | $77.67_{\pm2.05}$ | $89.31_{\pm0.61}$ | $60.58_{\pm0.30}$ | $71.14_{\pm9.40}$ | 5.63 |
| | DyGMoCE | $85.62_{\pm0.37}$ | $\mathbf{84.84_{\pm0.09}}$ | $\mathbf{92.61_{\pm0.82}}$ | $\mathbf{86.03_{\pm1.62}}$ | $\mathbf{83.15_{\pm0.47}}$ | $\mathbf{97.12_{\pm1.04}}$ | $\mathbf{91.79_{\pm0.63}}$ | $\mathbf{79.36_{\pm0.82}}$ | 1.25 |
| ind | TGAT | $81.93_{\pm0.22}$ | $87.13_{\pm0.20}$ | $73.18_{\pm0.33}$ | $63.99_{\pm0.21}$ | $60.45_{\pm2.12}$ | $71.62_{\pm5.42}$ | $66.13_{\pm0.78}$ | $53.04_{\pm2.58}$ | 6.25 |
| | GraphMixer | $84.28_{\pm0.30}$ | $82.21_{\pm0.13}$ | $72.45_{\pm0.72}$ | $60.22_{\pm0.32}$ | $71.53_{\pm0.85}$ | $84.22_{\pm0.91}$ | $66.53_{\pm1.22}$ | $51.89_{\pm0.74}$ | 6.50 |
| | DyGFormer | $75.09_{\pm3.70}$ | $86.23_{\pm0.51}$ | $80.76_{\pm0.76}$ | $69.25_{\pm0.36}$ | $74.07_{\pm0.64}$ | $87.96_{\pm1.80}$ | $62.56_{\pm1.51}$ | $53.37_{\pm1.26}$ | 5.00 |
| | RepeatMixer | $84.52_{\pm0.64}$ | $87.71_{\pm0.33}$ | $80.41_{\pm5.06}$ | $47.57_{\pm13.6}$ | $59.82_{\pm14.9}$ | $87.14_{\pm0.07}$ | $54.04_{\pm17.1}$ | $44.66_{\pm1.35}$ | 6.25 |
| | TPNet | $77.56_{\pm0.39}$ | $82.20_{\pm0.27}$ | $\underline{85.97_{\pm1.58}}$ | $72.99_{\pm1.63}$ | $\underline{77.54_{\pm1.28}}$ | $\underline{94.73_{\pm0.64}}$ | $\underline{90.14_{\pm0.41}}$ | $\underline{78.38_{\pm0.41}}$ | 3.63 |
| | STGN$^h$ | $80.16_{\pm0.14}$ | $86.24_{\pm0.07}$ | $76.87_{\pm0.29}$ | $64.73_{\pm0.17}$ | $65.45_{\pm1.31}$ | $79.08_{\pm1.19}$ | $70.31_{\pm0.64}$ | $59.87_{\pm1.65}$ | 5.38 |
| | TIDFormer | $\underline{84.54_{\pm0.50}}$ | $\mathbf{89.67_{\pm5.78}}$ | $82.29_{\pm0.78}$ | $\mathbf{78.46_{\pm0.98}}$ | $76.68_{\pm1.25}$ | $62.14_{\pm6.73}$ | $48.99_{\pm2.27}$ | $37.16_{\pm3.62}$ | 4.63 |
| | TAMI | $66.90_{\pm5.85}$ | $84.09_{\pm1.14}$ | $76.17_{\pm1.22}$ | $61.94_{\pm0.73}$ | $72.83_{\pm0.82}$ | $87.81_{\pm1.82}$ | $59.33_{\pm1.83}$ | $71.27_{\pm1.40}$ | 6.13 |
| | DyGMoCE | $84.69_{\pm0.48}$ | $\underline{88.75_{\pm0.32}}$ | $\mathbf{87.68_{\pm0.41}}$ | $\underline{76.87_{\pm1.26}}$ | $\mathbf{78.96_{\pm0.94}}$ | $\mathbf{96.28_{\pm0.53}}$ | $\mathbf{92.49_{\pm0.64}}$ | $\mathbf{80.85_{\pm1.02}}$ | 1.25 |

*Table 8.* AP for inductive dynamic link prediction with random, historical, and inductive negative sampling strategies (NSS). The best and runner-up results are emphasized by **bold** and underlined fonts. Rank indicates the average rank across datasets.

| NSS | Methods | Wikipedia | Reddit | MOOC | LastFM | Enron | US Legis. | UN Trade | UN Vote | Rank |
|---|---|---|---|---|---|---|---|---|---|---|
| rnd | TGAT | $96.22_{\pm 0.07}$ | $97.09_{\pm 0.04}$ | $85.50_{\pm 0.19}$ | $78.63_{\pm 0.31}$ | $67.05_{\pm 1.51}$ | $51.00_{\pm 3.11}$ | $61.03_{\pm 0.18}$ | $52.24_{\pm 1.46}$ | 7.75 |
| | GraphMixer | $96.65_{\pm 0.02}$ | $95.26_{\pm 0.02}$ | $81.41_{\pm 0.21}$ | $82.11_{\pm 0.42}$ | $75.88_{\pm 0.48}$ | $50.71_{\pm 0.76}$ | $62.17_{\pm 0.31}$ | $50.68_{\pm 0.44}$ | 8.13 |
| | DyGFormer | $98.59_{\pm 0.03}$ | $\underline{98.84}_{\pm 0.02}$ | $86.96_{\pm 0.43}$ | $94.23_{\pm 0.09}$ | $89.76_{\pm 0.34}$ | $54.28_{\pm 2.87}$ | $64.55_{\pm 0.62}$ | $55.93_{\pm 0.39}$ | 3.94 |
| | RepeatMixer | $98.56_{\pm 0.06}$ | $98.59_{\pm 0.02}$ | $84.20_{\pm 0.09}$ | $92.89_{\pm 0.15}$ | $87.91_{\pm 0.30}$ | $49.03_{\pm 0.71}$ | $60.93_{\pm 0.23}$ | $54.51_{\pm 0.15}$ | 6.88 |
| | TPNet | $98.89_{\pm 0.03}$ | $98.79_{\pm 0.02}$ | $\underline{93.63}_{\pm 0.15}$ | $\underline{95.47}_{\pm 0.09}$ | $90.34_{\pm 0.28}$ | $\underline{61.53}_{\pm 0.03}$ | $\underline{85.94}_{\pm 0.06}$ | $51.22_{\pm 0.48}$ | $\underline{3.25}$ |
| | STGN$^h$ | $98.59_{\pm 0.06}$ | $98.72_{\pm 0.02}$ | $88.74_{\pm 0.25}$ | $91.64_{\pm 0.37}$ | $83.28_{\pm 0.78}$ | $55.62_{\pm 2.45}$ | $63.98_{\pm 0.13}$ | $55.18_{\pm 1.10}$ | 5.06 |
| | TIDFormer | $98.86_{\pm 0.01}$ | $98.77_{\pm 0.16}$ | $91.50_{\pm 0.28}$ | $94.61_{\pm 0.70}$ | $85.70_{\pm 0.75}$ | $51.81_{\pm 0.23}$ | $50.85_{\pm 1.91}$ | $\mathbf{62.10}_{\pm 1.29}$ | 4.50 |
| | TAMI | $\underline{98.91}_{\pm 0.01}$ | $98.83_{\pm 0.01}$ | $86.39_{\pm 0.29}$ | $94.53_{\pm 0.11}$ | $\underline{90.35}_{\pm 0.31}$ | $50.79_{\pm 0.03}$ | $62.39_{\pm 0.70}$ | $54.45_{\pm 2.04}$ | 4.38 |
| | DyGMoCE | $\mathbf{98.96}_{\pm 0.04}$ | $\mathbf{99.01}_{\pm 0.02}$ | $\mathbf{94.25}_{\pm 0.33}$ | $\mathbf{96.74}_{\pm 0.48}$ | $\mathbf{95.96}_{\pm 0.12}$ | $\mathbf{63.87}_{\pm 1.06}$ | $\mathbf{87.02}_{\pm 0.06}$ | $\underline{56.74}_{\pm 0.61}$ | **1.13** |
| hist | TGAT | $84.17_{\pm 0.22}$ | $63.47_{\pm 0.36}$ | $76.73_{\pm 0.29}$ | $76.27_{\pm 0.25}$ | $61.40_{\pm 1.31}$ | $51.83_{\pm 3.95}$ | $55.28_{\pm 0.71}$ | $53.05_{\pm 3.10}$ | 6.25 |
| | GraphMixer | $\mathbf{87.60}_{\pm 0.30}$ | $64.50_{\pm 0.26}$ | $74.00_{\pm 0.97}$ | $76.42_{\pm 0.22}$ | $\underline{72.37}_{\pm 1.37}$ | $52.03_{\pm 1.02}$ | $54.94_{\pm 0.97}$ | $48.09_{\pm 0.43}$ | 5.25 |
| | DyGFormer | $71.42_{\pm 4.43}$ | $65.37_{\pm 0.60}$ | $80.82_{\pm 0.30}$ | $76.35_{\pm 0.52}$ | $67.07_{\pm 0.62}$ | $56.31_{\pm 3.46}$ | $53.20_{\pm 1.07}$ | $52.63_{\pm 1.26}$ | 5.50 |
| | RepeatMixer | $\underline{86.45}_{\pm 0.52}$ | $\mathbf{68.16}_{\pm 1.66}$ | $\underline{84.69}_{\pm 4.22}$ | $68.30_{\pm 14.20}$ | $66.58_{\pm 11.30}$ | $51.45_{\pm 2.73}$ | $56.12_{\pm 10.70}$ | $45.41_{\pm 1.02}$ | 5.38 |
| | TPNet | $74.40_{\pm 0.91}$ | $60.79_{\pm 0.29}$ | $83.54_{\pm 1.81}$ | $\underline{82.17}_{\pm 1.93}$ | $71.39_{\pm 0.31}$ | $\underline{66.25}_{\pm 2.02}$ | $\mathbf{78.02}_{\pm 0.03}$ | $62.43_{\pm 0.42}$ | $\underline{3.75}$ |
| | STGN$^h$ | $83.52_{\pm 0.18}$ | $64.33_{\pm 0.29}$ | $80.04_{\pm 0.23}$ | $78.84_{\pm 0.20}$ | $65.71_{\pm 1.05}$ | $55.92_{\pm 2.16}$ | $58.14_{\pm 0.57}$ | $55.11_{\pm 1.48}$ | 4.63 |
| | TIDFormer | $71.68_{\pm 4.35}$ | $63.45_{\pm 1.46}$ | $76.33_{\pm 1.55}$ | $77.14_{\pm 4.08}$ | $71.97_{\pm 2.24}$ | $53.28_{\pm 4.75}$ | $49.67_{\pm 0.73}$ | $46.88_{\pm 0.99}$ | 6.50 |
| | TAMI | $61.12_{\pm 9.26}$ | $64.87_{\pm 1.09}$ | $75.73_{\pm 1.48}$ | $70.65_{\pm 1.70}$ | $68.33_{\pm 0.65}$ | $57.19_{\pm 0.65}$ | $51.10_{\pm 0.07}$ | $54.13_{\pm 0.83}$ | 6.13 |
| | DyGMoCE | $81.83_{\pm 1.34}$ | $\underline{65.72}_{\pm 0.79}$ | $\mathbf{85.18}_{\pm 0.94}$ | $\mathbf{83.61}_{\pm 1.27}$ | $\mathbf{72.50}_{\pm 0.26}$ | $\mathbf{68.42}_{\pm 1.92}$ | $\underline{81.21}_{\pm 0.16}$ | $\mathbf{64.93}_{\pm 0.85}$ | **1.63** |
| ind | TGAT | $84.17_{\pm 0.22}$ | $63.40_{\pm 0.36}$ | $76.72_{\pm 0.30}$ | $76.28_{\pm 0.25}$ | $61.40_{\pm 1.30}$ | $51.83_{\pm 3.95}$ | $55.58_{\pm 0.68}$ | $53.08_{\pm 3.10}$ | 6.50 |
| | GraphMixer | $\mathbf{87.60}_{\pm 0.29}$ | $64.49_{\pm 0.25}$ | $73.99_{\pm 0.97}$ | $76.42_{\pm 0.22}$ | $\underline{72.37}_{\pm 1.38}$ | $52.03_{\pm 1.02}$ | $54.88_{\pm 1.01}$ | $48.10_{\pm 0.40}$ | 5.38 |
| | DyGFormer | $71.42_{\pm 4.43}$ | $65.35_{\pm 0.60}$ | $80.82_{\pm 0.30}$ | $76.35_{\pm 0.52}$ | $67.07_{\pm 0.62}$ | $56.31_{\pm 3.46}$ | $52.56_{\pm 1.70}$ | $52.61_{\pm 1.25}$ | 5.50 |
| | RepeatMixer | $\underline{86.45}_{\pm 0.52}$ | $\underline{68.16}_{\pm 1.66}$ | $\underline{84.69}_{\pm 4.23}$ | $68.30_{\pm 14.30}$ | $66.60_{\pm 11.30}$ | $51.45_{\pm 2.73}$ | $56.13_{\pm 10.70}$ | $45.41_{\pm 1.00}$ | 5.38 |
| | TPNet | $74.40_{\pm 0.92}$ | $60.79_{\pm 0.28}$ | $83.54_{\pm 1.81}$ | $82.17_{\pm 1.93}$ | $71.38_{\pm 0.31}$ | $\underline{66.25}_{\pm 2.02}$ | $\underline{78.14}_{\pm 0.01}$ | $\underline{62.50}_{\pm 0.44}$ | $\underline{3.88}$ |
| | STGN$^h$ | $85.42_{\pm 0.28}$ | $65.12_{\pm 0.29}$ | $78.94_{\pm 0.24}$ | $79.36_{\pm 0.20}$ | $65.85_{\pm 1.04}$ | $56.05_{\pm 2.16}$ | $58.64_{\pm 0.54}$ | $56.93_{\pm 1.48}$ | 4.38 |
| | TIDFormer | $79.24_{\pm 3.31}$ | $\mathbf{69.70}_{\pm 1.59}$ | $78.49_{\pm 1.77}$ | $\underline{82.81}_{\pm 2.30}$ | $70.70_{\pm 1.85}$ | $53.28_{\pm 4.75}$ | $49.52_{\pm 1.06}$ | $46.07_{\pm 0.94}$ | 5.25 |
| | TAMI | $61.47_{\pm 7.00}$ | $63.37_{\pm 0.32}$ | $76.58_{\pm 0.73}$ | $65.76_{\pm 1.45}$ | $67.76_{\pm 0.55}$ | $55.00_{\pm 0.02}$ | $54.18_{\pm 0.53}$ | $55.98_{\pm 0.47}$ | 6.88 |
| | DyGMoCE | $80.81_{\pm 0.63}$ | $65.22_{\pm 0.41}$ | $\mathbf{85.90}_{\pm 1.86}$ | $\mathbf{83.39}_{\pm 0.84}$ | $\mathbf{73.78}_{\pm 0.49}$ | $\mathbf{68.79}_{\pm 1.13}$ | $\mathbf{80.46}_{\pm 1.48}$ | $\mathbf{65.35}_{\pm 0.72}$ | **1.88** |

*Table 9.* AUC-ROC for inductive dynamic link prediction with random, historical, and inductive negative sampling strategies (NSS). The best and runner-up results are emphasized by **bold** and underlined fonts. Rank indicates the average rank across datasets.

| NSS | Methods | Wikipedia | Reddit | MOOC | LastFM | Enron | US Legis. | UN Trade | UN Vote | Rank |
|---|---|---|---|---|---|---|---|---|---|---|
| rnd | TGAT | $95.90_{\pm 0.09}$ | $96.98_{\pm 0.04}$ | $86.84_{\pm 0.17}$ | $76.99_{\pm 0.29}$ | $64.63_{\pm 1.74}$ | $48.27_{\pm 3.50}$ | $62.72_{\pm 0.12}$ | $51.83_{\pm 1.35}$ | 8.13 |
| | GraphMixer | $96.30_{\pm 0.04}$ | $94.97_{\pm 0.05}$ | $82.77_{\pm 0.24}$ | $80.37_{\pm 0.18}$ | $76.51_{\pm 0.71}$ | $47.20_{\pm 0.89}$ | $63.48_{\pm 0.37}$ | $50.04_{\pm 0.86}$ | 8.13 |
| | DyGFormer | $98.48_{\pm 0.03}$ | $\underline{98.71}_{\pm 0.01}$ | $87.62_{\pm 0.51}$ | $94.08_{\pm 0.08}$ | $90.69_{\pm 0.26}$ | $53.21_{\pm 3.04}$ | $67.25_{\pm 1.05}$ | $56.73_{\pm 0.69}$ | 3.63 |
| | RepeatMixer | $98.44_{\pm 0.04}$ | $98.38_{\pm 0.03}$ | $84.81_{\pm 0.09}$ | $92.67_{\pm 0.04}$ | $88.91_{\pm 0.08}$ | $47.71_{\pm 1.19}$ | $65.67_{\pm 0.40}$ | $54.54_{\pm 0.34}$ | 6.38 |
| | TPNet | $\underline{98.90}_{\pm 0.03}$ | $98.65_{\pm 0.02}$ | $\underline{94.35}_{\pm 0.14}$ | $\underline{95.44}_{\pm 0.07}$ | $90.40_{\pm 0.65}$ | $\underline{64.37}_{\pm 0.44}$ | $\underline{86.60}_{\pm 0.07}$ | $53.77_{\pm 2.65}$ | $\underline{3.25}$ |
| | STGN$^h$ | $97.72_{\pm 0.07}$ | $98.18_{\pm 0.03}$ | $89.35_{\pm 0.14}$ | $80.14_{\pm 0.23}$ | $70.85_{\pm 1.39}$ | $50.46_{\pm 2.81}$ | $66.98_{\pm 1.02}$ | $54.62_{\pm 1.08}$ | 5.88 |
| | TIDFormer | $98.84_{\pm 0.01}$ | $98.66_{\pm 0.18}$ | $92.38_{\pm 0.23}$ | $94.17_{\pm 0.69}$ | $86.47_{\pm 0.52}$ | $48.62_{\pm 1.07}$ | $51.31_{\pm 2.93}$ | $\mathbf{61.59}_{\pm 1.43}$ | 4.50 |
| | TAMI | $98.85_{\pm 0.01}$ | $98.70_{\pm 0.01}$ | $87.23_{\pm 0.37}$ | $94.25_{\pm 0.06}$ | $\underline{90.81}_{\pm 0.40}$ | $48.30_{\pm 0.14}$ | $66.04_{\pm 0.45}$ | $\underline{57.19}_{\pm 2.49}$ | 3.75 |
| | DyGMoCE | $\mathbf{99.15}_{\pm 0.03}$ | $\mathbf{98.89}_{\pm 0.03}$ | $\mathbf{95.81}_{\pm 0.10}$ | $\mathbf{96.27}_{\pm 0.05}$ | $\mathbf{91.26}_{\pm 0.54}$ | $\mathbf{67.19}_{\pm 1.40}$ | $\mathbf{88.05}_{\pm 0.29}$ | $55.64_{\pm 1.31}$ | **1.38** |
| hist | TGAT | $78.38_{\pm 0.20}$ | $64.43_{\pm 0.27}$ | $74.08_{\pm 0.27}$ | $69.89_{\pm 0.28}$ | $57.84_{\pm 2.18}$ | $49.99_{\pm 4.88}$ | $59.74_{\pm 0.59}$ | $51.73_{\pm 4.12}$ | 6.50 |
| | GraphMixer | $\mathbf{82.87}_{\pm 0.21}$ | $64.27_{\pm 0.13}$ | $72.53_{\pm 0.84}$ | $70.07_{\pm 0.20}$ | $68.20_{\pm 1.62}$ | $49.28_{\pm 0.86}$ | $59.88_{\pm 1.17}$ | $45.49_{\pm 0.42}$ | 5.63 |
| | DyGFormer | $68.33_{\pm 2.82}$ | $64.81_{\pm 0.25}$ | $80.77_{\pm 0.63}$ | $70.73_{\pm 0.37}$ | $65.78_{\pm 0.42}$ | $56.57_{\pm 3.22}$ | $58.46_{\pm 1.65}$ | $53.85_{\pm 2.02}$ | 4.88 |
| | RepeatMixer | $79.06_{\pm 0.54}$ | $\mathbf{66.93}_{\pm 0.82}$ | $81.13_{\pm 5.08}$ | $61.62_{\pm 13.3}$ | $63.56_{\pm 12.1}$ | $48.77_{\pm 2.11}$ | $53.87_{\pm 14.6}$ | $35.90_{\pm 1.54}$ | 6.25 |
| | TPNet | $70.42_{\pm 0.29}$ | $62.14_{\pm 0.05}$ | $83.85_{\pm 1.03}$ | $\underline{77.18}_{\pm 1.00}$ | $70.74_{\pm 0.45}$ | $66.86_{\pm 1.51}$ | $80.37_{\pm 0.21}$ | $60.29_{\pm 0.33}$ | $\underline{3.50}$ |
| | STGN$^h$ | $\underline{80.92}_{\pm 0.16}$ | $65.05_{\pm 0.28}$ | $77.41_{\pm 0.22}$ | $70.64_{\pm 0.73}$ | $61.33_{\pm 1.47}$ | $52.65_{\pm 2.90}$ | $62.19_{\pm 0.44}$ | $53.42_{\pm 3.06}$ | 4.63 |
| | TIDFormer | $69.56_{\pm 2.26}$ | $63.40_{\pm 0.81}$ | $77.63_{\pm 0.83}$ | $72.29_{\pm 3.47}$ | $\underline{71.43}_{\pm 1.78}$ | $50.92_{\pm 8.49}$ | $49.24_{\pm 1.70}$ | $42.20_{\pm 2.53}$ | 5.88 |
| | TAMI | $61.46_{\pm 7.00}$ | $63.37_{\pm 0.32}$ | $76.57_{\pm 0.73}$ | $65.76_{\pm 1.45}$ | $67.76_{\pm 0.55}$ | $55.00_{\pm 0.02}$ | $54.24_{\pm 0.56}$ | $56.02_{\pm 0.44}$ | 6.38 |
| | DyGMoCE | $79.15_{\pm 0.39}$ | $\underline{65.77}_{\pm 0.13}$ | $\mathbf{84.76}_{\pm 0.84}$ | $\mathbf{79.53}_{\pm 0.65}$ | $\mathbf{72.19}_{\pm 0.27}$ | $\mathbf{69.15}_{\pm 1.81}$ | $\mathbf{83.02}_{\pm 0.76}$ | $\mathbf{63.26}_{\pm 0.93}$ | **1.38** |
| ind | TGAT | $78.38_{\pm 0.20}$ | $64.39_{\pm 0.27}$ | $74.07_{\pm 0.27}$ | $69.89_{\pm 0.28}$ | $57.83_{\pm 2.18}$ | $49.99_{\pm 4.88}$ | $59.98_{\pm 0.59}$ | $51.78_{\pm 4.14}$ | 6.63 |
| | GraphMixer | $\mathbf{82.88}_{\pm 0.21}$ | $64.27_{\pm 0.13}$ | $72.52_{\pm 0.84}$ | $70.07_{\pm 0.20}$ | $68.19_{\pm 1.63}$ | $49.28_{\pm 0.86}$ | $59.71_{\pm 1.17}$ | $45.57_{\pm 0.41}$ | 5.88 |
| | DyGFormer | $68.33_{\pm 2.82}$ | $64.80_{\pm 0.25}$ | $80.77_{\pm 0.63}$ | $70.73_{\pm 0.37}$ | $65.79_{\pm 0.42}$ | $56.57_{\pm 3.22}$ | $57.28_{\pm 3.06}$ | $53.87_{\pm 2.01}$ | 5.38 |
| | RepeatMixer | $79.06_{\pm 0.54}$ | $66.93_{\pm 0.82}$ | $81.13_{\pm 5.08}$ | $61.68_{\pm 13.3}$ | $63.57_{\pm 12.1}$ | $48.77_{\pm 2.11}$ | $53.84_{\pm 14.7}$ | $35.90_{\pm 1.53}$ | 6.75 |
| | TPNet | $70.43_{\pm 0.30}$ | $62.13_{\pm 0.04}$ | $\underline{83.85}_{\pm 1.03}$ | $77.18_{\pm 1.00}$ | $\underline{70.73}_{\pm 0.45}$ | $66.86_{\pm 1.51}$ | $80.47_{\pm 0.18}$ | $60.38_{\pm 0.37}$ | $\underline{3.63}$ |
| | STGN$^h$ | $\underline{80.50}_{\pm 0.26}$ | $65.68_{\pm 0.37}$ | $78.43_{\pm 0.63}$ | $72.25_{\pm 1.42}$ | $64.72_{\pm 0.69}$ | $53.61_{\pm 3.94}$ | $61.67_{\pm 1.48}$ | $54.03_{\pm 1.59}$ | 4.38 |
| | TIDFormer | $78.50_{\pm 1.56}$ | $\mathbf{70.76}_{\pm 0.64}$ | $81.16_{\pm 0.80}$ | $\mathbf{79.36}_{\pm 1.29}$ | $69.75_{\pm 1.79}$ | $50.92_{\pm 8.49}$ | $48.87_{\pm 2.25}$ | $41.50_{\pm 2.52}$ | 4.50 |
| | TAMI | $61.47_{\pm 7.00}$ | $63.37_{\pm 0.32}$ | $76.58_{\pm 0.73}$ | $65.76_{\pm 1.45}$ | $67.76_{\pm 0.55}$ | $55.00_{\pm 0.02}$ | $54.18_{\pm 0.53}$ | $55.98_{\pm 0.47}$ | 6.38 |
| | DyGMoCE | $80.34_{\pm 0.58}$ | $\underline{67.30}_{\pm 0.29}$ | $\mathbf{84.21}_{\pm 0.88}$ | $\underline{78.67}_{\pm 1.15}$ | $\mathbf{70.93}_{\pm 0.80}$ | $\mathbf{68.92}_{\pm 0.69}$ | $\mathbf{81.66}_{\pm 0.26}$ | $\mathbf{62.78}_{\pm 1.64}$ | **1.50** |

*Table 10.* Training time and memory usage of DyGMoCE with Euclidean attention (EA), Riemannian attention (RA), and our fast Riemannian attention (Fast RA). Reduction ratios (↓%) show the relative time and memory savings of fast Riemannian attention over Riemannian attention. $L$ is the input length. OOM stands for Out-Of-Memory.

| Datasets | $L$ | Running Time | | | | Memory Usage | | | |
|---|---|---|---|---|---|---|---|---|---|
| | | EA | RA | Fast RA | ↓ | EA | RA | Fast RA | ↓ |
| Wikipedia | 16 | 2 min 07 s | 4 min 17 s | 3 min 06 s | 28.7% | 2,820 MB | 7,434 MB | 4,762 MB | 35.9% |
| | 32 | 2 min 27 s | 5 min 26 s | 3 min 40 s | 32.5% | 5,418 MB | 20,112 MB | 9,350 MB | 53.5% |
| | 64 | 3 min 15 s | — | 5 min 40 s | — | 10,524 MB | OOM | 18,468 MB | — |
| Reddit | 16 | 7 min 06 s | 13 min 43 s | 11 min 20 s | 17.3% | 3,212 MB | 7,795 MB | 5,149 MB | 33.9% |
| | 32 | 8 min 45 s | 19 min 23 s | 14 min 31 s | 25.1% | 5,791 MB | 20,455 MB | 9,752 MB | 52.3% |
| | 64 | 13 min 18 s | — | — | — | 10,911 MB | OOM | OOM | — |
| LastFM | 16 | 11 min 27 s | 24 min 53 s | 18 min 13 s | 26.8% | 3,645 MB | 8,239 MB | 5,576 MB | 32.3% |
| | 32 | 15 min 05 s | 34 min 27 s | 23 min 50 s | 30.8% | 6,226 MB | 2,0928 MB | 10,195 MB | 51.3% |
| | 64 | 22 min 48 s | — | 41 min 15 s | — | 11,331 MB | OOM | 19,264 MB | — |
| Enron | 16 | 2 min 15 s | 3 min 54 s | 3 min 11 s | 18.4% | 2,787 MB | 7,377 MB | 4,717 MB | 36.1% |
| | 32 | 2 min 30 s | 5 min 04 s | 3 min 32 s | 30.3% | 5,368 MB | 20,044 MB | 9,313 MB | 53.5% |
| | 64 | 3 min 28 s | — | 6 min 11 s | — | 10,468 MB | OOM | 18,438 MB | — |
| US Legis. | 16 | 2 min 01 s | 3 min 10 s | 3 min 07 s | 1.6% | 2,747 MB | 7,325 MB | 4,684 MB | 36.1% |
| | 32 | 1 min 12 s | 3 min 54 s | 3 min 15 s | 16.7% | 5,333 MB | 20,013 MB | 9,276 MB | 53.7% |
| | 64 | 2 min 45 s | — | 3 min 18 s | — | 10,426 MB | OOM | 18,336 MB | — |
| UN Trade | 16 | 6 min 07 s | 10 min 36 s | 9 min 55 s | 6.4% | 3,055 MB | 7,652 MB | 5,009 MB | 34.5% |
| | 32 | 8 min 24 s | 15 min 57 s | 11 min 54 s | 22.6% | 5,638 MB | 20,341 MB | 9,607 MB | 52.8% |
| | 64 | 11 min 06 s | — | 17 min 44 s | — | 10,775 MB | OOM | 18,712 MB | — |

