# OpenReview forum: "Fast Mixture of Curvature-Aware Experts for Diverse and Dynamic Graph Topologies"
_ICML.cc/2026/Conference — ICML 2026 regular_

### Official Review · Reviewer_R4Kq · 2026-02-28

**Soundness:** 4
**Presentation:** 3
**Significance:** 3
**Originality:** 4
**Overall Recommendation:** 5
**Confidence:** 5

**Summary:**

This paper proposes DyGMoCE, a novel dynamic graph Transformer with a mixture of curvature-aware experts that embeds each node at every timestamp into an adaptive curvature space. Both the attention and feed-forward modules consist of Riemannian experts with distinct curvatures. Since these experts are heterogeneous and exhibit continuously varying geometric properties, the paper introduces a Geometrically Continuous Expert Router (GCRouter) to adaptively select appropriate experts for each node. To improve efficiency, the authors reformulate Riemannian attention into a mathematically equivalent fast version using polar decomposition and the Riemannian law of cosines. Extensive experiments demonstrate that DyGMoCE outperforms existing SOTA methods on dynamic link prediction and node classification, while achieving a 26.3% speedup and a 52.0% reduction in memory usage.

**Compliance With Llm Reviewing Policy:**

Affirmed.

**Final Justification:**

The paper proposes a novel dynamic graph model with a mixture of curvature-aware experts. Its main strengths lie in the originality of the mixed-curvature design, the development of fast Riemannian attention, and the thorough experimental evaluation across multiple settings. My initial concerns mainly centered on the rationale for curvature initialization, whether fast Riemannian attention is mathematically equivalent to standard Riemannian attention, and how the zero-curvature case should be interpreted. After reading the rebuttal, I find these concerns adequately addressed through technical explanations and references. Therefore, I raise my recommendation from weak accept to accept.

**Key Questions For Authors:**

This paper presents a promising approach, but I still have the following questions:
1. How does DyGMoCE determine or initialize the curvature values for the hyperbolic and spherical models?
2. Is your proposed approach essentially an approximate optimization of Riemannian attention?
3. In Eq. (16), the formulation is provided for curvature values less than 0 and greater than 0. How is the case of zero curvature handled? Can it be unified within the same formulation, or is the standard Euclidean distance used instead?

**Limitations:**

yes

**Strengths And Weaknesses:**

Strengths:
1. The paper exhibits solid novelty. It observes that node neighborhoods in dynamic graphs may exhibit different locally optimal geometries across nodes and over time. To address this, the method adopts a Mixture-of-Experts framework with experts operating under different curvatures. Furthermore, GCRouter employs a sliding-window mechanism to select a contiguous block of Riemannian experts, enabling adaptive modeling of heterogeneous geometries with continuously varying curvature (from hyperbolic to Euclidean to spherical).
2. The paper is theoretically rigorous and demonstrates practical value for real-world dynamic graph applications. Given that continuous-time dynamic graphs require processing each interaction, the paper introduces a mathematically equivalent formulation of Fast Riemannian Attention that greatly improves computational efficiency and reduces memory usage.
3. The experimental results are thorough and convincing. For dynamic link prediction, the paper reports AP and AUC-ROC under both transductive and inductive settings with three negative sampling strategies. It further provides comprehensive empirical analyses that validate the effectiveness of each component, the model’s efficiency and robustness, and the advantages of combining experts with diverse type and curvature configurations.
4. The code is released, supporting the reproducibility.
5. The paper presents a thorough analysis of the computational complexity and FLOPs of Euclidean, Riemannian, and Fast Riemannian attention in dynamic graphs.

Weaknesses
1. The manuscript contains several grammatical errors, and the authors should carefully proofread and revise them. For example, in the Related Work section, “a shared hyperbolic spaces” should be “a shared hyperbolic space,” and “which align with our setting” should be “which aligns with our setting.”
2. The paper dedicates excessive space to technical details, which may compromise readability for readers unfamiliar with Riemannian geometry. Including pseudocode would help clarify the algorithmic flow and improve accessibility.
3. To further enhance reproducibility beyond the provided code and search ranges, the authors should include the specific optimal hyperparameter configurations for each dataset in the released codebase.
4. The manuscript employs a wide variety of mathematical notations. It is recommended that the article include a table of symbols to explicitly define the meaning of each symbol.

---

> ### Author Rebuttal · Authors · 2026-03-31
>
> We sincerely thank the reviewer for the time and effort in reviewing our paper. We take all comments seriously and try our best to address every raised concern. We have incorporated the corresponding revisions into the updated manuscript. Please feel free to ask any follow-up questions.
>
> ## W1
>
> Thank you for your careful checks. As suggested by the reviewer, we have corrected the "a shared hyperbolic spaces" to "a shared hyperbolic space", and "which align with our setting" to "which aligns with our setting". We have carefully reviewed and revised the manuscript to correct other grammatical issues as well.
>
> ## W2
>
> Thank you for your suggestion. We have revised the manuscript to include pseudocode, which clarifies the algorithmic flow and enhances accessibility for readers unfamiliar with Riemannian geometry. **The pseudocode is shown at** https://postimg.cc/gw7Q9ZjR.
>
> ## W3
>
> We thank the reviewer for the valuable suggestion. We will include the optimal hyperparameter configurations for each dataset in the released codebase to further enhance reproducibility. **The optimal hyperparameter configurations are shown in** https://postimg.cc/xXR5nRbL.
>
> ## W4
>
> We thank the reviewer for the suggestion. We will add a comprehensive table of symbols in the revised manuscript to clearly define all notations used. **The table of symbols is shown in** https://postimg.cc/d7VB1y87.
>
>
>
>
> ## Q1
>
> We thank the reviewer for this insightful question. We initialize the curvature values by jointly considering **numerical stability, curvature diversity, and empirical performance**.
>
> **Numerical stability.** HGCN [1] notes that when curvature is close to zero or excessively large, limited machine precision may cause substantial rounding errors. We therefore restrict curvature values to a reasonable range.
>
> **Curvature diversity.** Some prior works [1,2] initialize learnable curvature as ($\kappa=-1$). However, ACE-HGNN [3] points out that curvature updates are typically small during training, making a single learnable curvature difficult to optimize effectively. Instead of relying on such a continuous parameter, we adopt an MoE framework with a discrete predefined curvature set ([-3,-2,-1,0,1,2,3]).
>
> **Empirical performance.** Section 5.5 shows that this setting provides the best overall trade-off between effectiveness and efficiency.
>
>
> ## Q2
>
> We thank the reviewer for highlighting this crucial point. We would like to clarify that our proposed Fast Riemannian Attention is not an approximate optimization; rather, it is a mathematically equivalent reformulation of traditional Riemannian attention. While many acceleration techniques for Transformers rely on numerical approximations to reduce computational overhead, our method achieves acceleration through a strictly analytical derivation.
>
> Specifically, we use the polar coordinate representation in Riemannian geometry to avoid costly vector-level operations, including Möbius addition and Möbius scalar multiplication. The equivalence is reflected in two aspects:
>
> **Similarity computation**.
> In standard Riemannian attention, geodesic distance is computed via Möbius addition. We instead apply the Riemannian law of cosines to the polar forms of the query and key, yielding exactly the same geodesic distance using only scalar trigonometric and hyperbolic functions.
>
> **Value aggregation**.
> By exploiting polar decomposition, we analytically combine Möbius scalar multiplication with the logarithmic map, leading to a simplified yet equivalent formulation of the aggregation process.
>
>
> ## Q3
>
> We thank the reviewer for this important question. The zero-curvature case does not require a separate distance definition; it is handled as the continuous limit of the same formulation. Specifically, as $\kappa \to 0$, Eq. (16) smoothly reduces to
> $ D_0(Q_i, K_j) = 2 \lVert Q_i - K_j \rVert .$ Therefore, under our definition, the $\kappa = 0$ case corresponds not to the standard Euclidean distance $\lVert Q_i - K_j \rVert$, but to its scaled form $2 \lVert Q_i - K_j \rVert$. This is also consistent with prior literature [2], regarding the behavior of the Riemannian distance as $\kappa \to 0$. We also provide a Taylor-expansion-based derivation of this Euclidean limit in Appendix C.2 (Riemannian Law of Cosines), and we will clarify this point more explicitly in the revised manuscript.
>
> ---
> References:
>
> [1] Hyperbolic Graph Convolutional Neural Networks, NeurIPS 2019.
>
> [2] Hyperbolic Graph Attention Network, IEEE TRANSACTIONS ON BIG DATA 2022.
>
> [3] ACE-HGNN: Adaptive Curvature Exploration Hyperbolic Graph Neural Network, ICDM 2021.

---

> > ### Author Rebuttal · Reviewer_R4Kq · 2026-04-02
> >
> > I have reviewed the authors' rebuttal and other reviewers' comments. All of my previous concerns have been satisfactorily addressed. The paper brings a good idea and new content to dynamic graph learning. I am pleased to raise my evaluation score to 5.

---

> > > ### Author Response · Authors · 2026-04-02
> > >
> > > Dear Reviewer R4Kq,
> > >
> > > Thank you for your positive feedback and for updating your rating after considering our rebuttal. We are glad to know that our rebuttal has addressed your concerns.
> > >
> > > As dynamic graphs evolve over time, local neighborhoods may exhibit diverse and time-varying topological patterns, making it difficult for a single geometry to model them adequately. To address this challenge, we propose DyGMoCE, a dynamic graph Transformer with a mixture of curvature-aware experts, which enables each node at each timestamp to adaptively embed in a mixed-curvature space. In particular, our method introduces curvature-aware experts into both the attention and FFN modules, designs a geometrically continuous router with a ranking constraint to better align expert selection with local topology, and further develops a fast Riemannian attention module to significantly improve computational efficiency.
> > >
> > >
> > > We sincerely appreciate your recognition of our work. In the revised version, we will further polish the presentation and ensure that the important clarifications from the rebuttal are clearly incorporated into the manuscript. Thank you again for your valuable time, insightful comments, and supportive evaluation.
> > >
> > > Best regards,
> > >
> > > Authors

---

### Official Review · Reviewer_aZgE · 2026-03-07

**Soundness:** 3
**Presentation:** 3
**Significance:** 2
**Originality:** 3
**Overall Recommendation:** 4
**Confidence:** 4

**Summary:**

This paper introduces a framework, DyGMoCE, a Dynamic Graph Transformer with a Mixture of Curvature-aware Experts for learning representations on continuous-time dynamic graphs. DyGMoCE adopts mixture-of-experts with Riemannian Transformers framework to learn mixed-curvature embedding spaces for each node at every timestamp. DyGMoCE also introduces fast attention version of the Riemannian Attention to speed up the training process and reduce memory usage.

**Compliance With Llm Reviewing Policy:**

Affirmed.

**Key Questions For Authors:**

Q1. The authors should provide a comparison between DyGMoCE’s performance and multi-hop neighbor sampling.

Q2. Experiments on larger benchmarks, such as TGB or TGB-Seq, are necessary to strengthen the paper’s contributions.

**Limitations:**

Yes

**Strengths And Weaknesses:**

**Strength:**

S1. The idea of using a mixture-of-experts to adaptively find the most suitable curvature for each node at different timestamps in dynamic graphs is interesting.

S2. The formulation of curvature mixing is mathematically consistent with Riemannian geometry constraints. DyGMoCE improve existing Riemannian Transformer/Attention with a light-weight version using Riemannian law of cosines and polar decomposition. Teh paper also proposes GCRouter, an improvement over standard MoE for selecting the best geometric expert.


**Weaknesses:**

W1. Hyperbolic geometry often shines in a hierarchical, multi-hop, tree-like structure. This has been shown in the static graph domain. However, DyGMoCE uses only the most recent 1-hop neighbors in dynamic graphs, which limits the claim of learning diverse topologies.

W2. While the paper argues that curvature diversity enhances expressiveness, it does not provide a theory showing that curvature mixing compensates for the lack of multi-hop propagation.

W3. Although the experiments are comprehensive, most datasets in the experiments are relatively small.  The author should
report DyGMoCE performance on large-scale benchmarks such as TGB
[1] or TGB-Seq [2]. Several reported baselines are not aligned with other works (TIDFormer[3], TPNet[4])

**References:**

1. Shenyang Huang, Farimah Poursafaei, Jacob Danovitch, Matthias Fey, Weihua Hu, Emanuele Rossi, Jure Leskovec, Michael Bronstein, Guillaume Rabusseau, and Reihaneh Rabbany. Temporal graph benchmark for machine
learning on temporal graphs. Advances in Neural Information Processing Systems.

2.  Lu Yi, Jie Peng, Yanping Zheng, Fengran Mo, Zhewei Wei, Yuhang Ye, Yue Zixuan, and Zengfeng Huang. TGB-Seq Benchmark: Challenging Temporal GNNs with Complex Sequential Dynamics. In Proceedings of the Thirteenth International Conference on Learning Representations, 2025.

3. Peng, Jie, Zhewei Wei, and Yuhang Ye. "TIDFormer: Exploiting Temporal and Interactive Dynamics Makes A Great Dynamic Graph Transformer." Proceedings of the 31st ACM SIGKDD Conference on Knowledge Discovery and Data Mining V. 2. 2025.

4. Xiaodong Lu, Leilei Sun, Tongyu Zhu, and Weifeng Lv. 2024. Improving Temporal Link Prediction via Temporal Walk Matrix Projection. arXiv preprint arXiv:2410.04013 (2024)

---

> ### Author Rebuttal · Authors · 2026-03-31
>
> We sincerely thank the reviewer for the time and effort in reviewing our paper. We take all comments seriously and try our best to address every raised concern. We have incorporated the corresponding revisions into the updated manuscript. Please feel free to ask any follow-up questions.
>
> ## W1 &  W2 & Q3
> We thank the reviewer for this valuable comment. We would like to clarify that DyGMoCE does not rely solely on the most recent 1-hop information. Instead, it explicitly incorporates higher-order and multi-hop structural information through two key mechanisms.
>
> **Higher-order interaction encoding $\mathbf{X}_I$**:  As described in Appendix B.2, we follow TPNet [1] and use a temporal walk matrix with walk length 3 to capture higher-order temporal paths and structural dependencies.
>
> **Higher-order graph-property encoding $\mathbf{X}_G$**: As described in Section 4.1, we characterize diverse local topologies using seven statistical summaries computed from each node’s 3-hop subgraph, including Gromov’s $\delta$-hyperbolicity, betweenness centrality, and clustering coefficient, among others.
>
> In constructing the input sequence $Seq_v^t$ for the Transformer, we collect each node together with its historical first-order interactions before time $t$. Following the reviewer’s suggestion, in the experiments below, we sample 32 recent 1-hop neighbors and 32 2-hop neighbors. The performance with multi-hop sampling shows a slight decrease, possibly due to the introduction of more noise. We leave the exploration of more suitable multi-hop sampling strategies to future work.
>
> |            | MOOC         | UN Trade     | UN Vote      |
> |------------|--------------|--------------|--------------|
> | DyGMoCE (1-hop {L=32})         | **96.83 ± 0.18**        | **90.20 ± 0.23**        | **78.24 ± 0.08**        |
> | DyGMoCE (1-hop&2-hop {L=32+32}) | 94.59 ± 0.25        | 85.41 ± 0.67        | 76.70 ± 0.16        |
>
> ## W3 & Q2
>
> We thank the reviewer for the valuable suggestions. Following the advice, we have conducted additional experiments on the TGB [2] and TGB-Seq [3] benchmarks. DyGMoCE achieves the best performance on these large-scale datasets, demonstrating its scalability and robustness.
> | TGB        | tgbl-wiki        | tgbl-review       | tgbl-coin         | tgbl-comment      |
> |------------|------------------|-------------------|-------------------|-------------------|
> | DyRep      | 5.0 ± 1.7        | 22.0 ± 3.0        | 45.2 ± 4.6        | 28.9 ± 3.3        |
> | TGN        | 39.6 ± 6.0       | 34.9 ± 2.0        | 58.6 ± 3.7        | 37.9 ± 2.1        |
> | DyGFormer  | 79.8 ± 0.4       | 22.4 ± 1.5        | 75.2 ± 0.4        | 67.0 ± 0.1        |
> | TPNet      | *82.7 ± 0.1*     | *41.8 ± 0.3*      | *83.2 ± 0.1*      | *82.5 ± 0.6*      |
> | DyGMoCE    | **85.3 ± 0.6**   | **45.2 ± 0.3**    | **85.7 ± 0.2**    | **84.2 ± 0.4**    |
>
>
>
> | TGB-Seq    | ML-20M       | Taobao      | Yelp         | GoogleLocal  |
> |------------|--------------|-------------|--------------|--------------|
> | JODIE      | 21.16 ± 0.73 | 48.36 ± 2.18 | *69.88 ± 0.31* | 41.86 ± 1.49 |
> | DyRep      | 19.00 ± 1.69 | 40.03 ± 2.40 | 57.69 ± 1.05 | 37.73 ± 1.34 |
> | TGAT       | 10.47 ± 0.20 | OOT         | OOT          | 19.78 ± 0.24 |
> | TGN        | *23.99 ± 0.20* | *60.28 ± 0.54* | 69.79 ± 0.24 | *54.13 ± 1.97* |
> | CAWN       | 12.31 ± 0.02 | OOT         | 25.71 ± 0.09 | 18.26 ± 0.02 |
> | TCL        | 12.04 ± 0.02 | 31.55 ± 0.03 | 24.39 ± 0.67 | 18.30 ± 0.02 |
> | GraphMixer | 21.97 ± 0.17 | 31.54 ± 0.02 | 33.96 ± 0.19 | 21.31 ± 0.14 |
> | DyGFormer  | OOT          | OOT         | 21.68 ± 0.20 | 18.39 ± 0.02 |
> | DyGMoCE    | **33.23 ± 0.45** | **71.74 ± 0.39** | **73.69 ± 0.68** | **62.56 ± 0.48** |
>
> ---
> We reproduced the baseline results using their official code and strictly followed the original hyperparameter settings. The slight performance variances from their published numbers likely result from different hardware environments.
>
> ---
> [1] Improving Temporal Link Prediction via Temporal Walk Matrix Projection, NeurIPS 2024.
>
> [2] Temporal Graph Benchmark for Machine Learning on Temporal Graphs, NeurIPS 2023.
>
> [3] TGB-Seq Benchmark: Challenging Temporal GNNs with Complex Sequential Dynamics, ICLR 2025.

---

> > ### Author Rebuttal · Reviewer_aZgE · 2026-04-01
> >
> > While I appreciate the authors’ efforts to address the concerns raised, I find that the rebuttal only partially addresses the weaknesses previously noted. Therefore, I have decided to retain my original score.

---

> > > ### Author Response · Authors · 2026-04-02
> > >
> > > Dear Reviewer aZgE,
> > >
> > > Thank you for your prompt response and for your positive recognition of our work. We are glad to know that our rebuttal and new experiments have partially addressed your concerns.
> > >
> > > As dynamic graphs evolve over time, local neighborhoods may exhibit diverse and time-varying topological patterns, making it difficult for a single geometry to model them adequately. To address this challenge, we propose DyGMoCE, a dynamic graph Transformer with a mixture of curvature-aware experts, which enables each node at each timestamp to adaptively embed in a mixed-curvature space. In particular, our method introduces curvature-aware experts into both the attention and FFN modules, designs a geometrically continuous router with a ranking constraint to better align expert selection with local topology, and further develops a fast Riemannian attention module to significantly improve computational efficiency.
> > >
> > >
> > > In our initial rebuttal, we primarily provided empirical analyses on large dynamic graph benchmarks (TGB and TGB-Seq) and evaluated the scalability and robustness of our method. We will incorporate these results in the revised version and further explore potential improvements from the perspectives you suggested. Thank you again for your valuable time, insightful comments, and supportive evaluation.
> > >
> > > Best regards,
> > >
> > > Authors

---

### Official Review · Reviewer_WEfN · 2026-03-08

**Soundness:** 3
**Presentation:** 2
**Significance:** 4
**Originality:** 4
**Overall Recommendation:** 3
**Confidence:** 4

**Summary:**

This paper focuses on the core problem of the diversity and temporal evolution of node neighborhood topologies in dynamic graphs. To address the limitation that existing methods embed graphs into a single curvature space, which fails to adapt to complex local topologies, it proposes DyGMoCE, a dynamic graph Transformer with a Mixture of Curvature-aware Experts. Its core contributions are: 1) Introducing curvature-aware Riemannian experts within both multi-head attention and FFN modules to construct mixed-curvature embedding spaces for each node at every timestamp. 2) Proposing a Geometrically Continuous Expert Router (GCRouter) that uses a sliding window to select a contiguous block of experts and aligns routing decisions with the local neighborhood's Gromov δ-hyperbolicity via a ranking loss. 3) Designing a Fast Riemannian Attention module that leverages polar decomposition and the Riemannian law of cosines to achieve mathematical equivalence with significant computational savings. Extensive experiments on 12 dynamic graph benchmarks demonstrate DyGMoCE's superior performance.

**Compliance With Llm Reviewing Policy:**

Affirmed.

**Final Justification:**

I thank the authors for their thoughtful response to the reviews. After carefully re-examining the original paper, the reviewers’ comments, and the authors’ rebuttal, I maintain my original score.

**Key Questions For Authors:**

1. Since this work involves hyperbolic, Euclidean, and spherical spaces, using Gromov's $\delta$-hyperbolicity—which is specifically tailored for hyperbolic geometry—may no longer be appropriate for representing node-level curvature. Have the authors considered utilizing discrete curvature measures such as Ricci curvature?
2. Are the curvature parameters $\kappa_i$ in the FFN and the Fast Riemannian Attention intended to be identical? While Figure 2 suggests they are the same, have the authors experimented with learning distinct curvature values for different modules?
3. Regarding the ''Riemannian FFN Experts'', is the Riemannian linear transformation implemented by first projecting the manifold data into Euclidean space for computation? If so, does this render the exponential and logarithmic mappings redundant in this context?
4. What is the underlying motivation for the operation used to calculate $\hat{\kappa}_v$ in Equation (12)? Is the intention to measure node curvature by taking a weighted sum of expert curvatures? Further clarification is needed as the current formulation is difficult to interpret.
5. The manuscript suffers from significant presentation issues, as several key elements lack adequate explanation. For instance, the circles and points in Figure 1 are not defined. Furthermore, notations such as $\hat{S}$, $\kappa_v^*$ are either undefined or not explained upon their first appearance, which severely hinders the coherence and readability of the paper.

**Limitations:**

yes

**Strengths And Weaknesses:**

Strengths:
1. The integration of a Mixture-of-Experts (MoE) framework with mixed-curvature Riemannian networks represents a novel and interesting approach to addressing the challenges of local topological diversity.
2. The proposed Fast Riemannian Attention module effectively addresses the computational efficiency issues inherent in DyGMoCE. Furthermore, its potential for integration into other sequence-based Transformer architectures demonstrates significant practical utility and strong generalizability.
3. The authors provide a thorough experimental evaluation across 12 datasets spanning various domains, such as social networks, transportation, and politics. The performance comparisons against eight state-of-the-art baselines for both dynamic link prediction and node classification tasks are highly convincing.

Weaknesses:
1. Since this work involves hyperbolic, Euclidean, and spherical spaces, using Gromov's $\delta$-hyperbolicity—which is specifically tailored for hyperbolic geometry—may no longer be appropriate for representing node-level curvature. Have the authors considered utilizing discrete curvature measures such as Ricci curvature?
2. Are the curvature parameters $\kappa_i$ in the FFN and the Fast Riemannian Attention intended to be identical? While Figure 2 suggests they are the same, have the authors experimented with learning distinct curvature values for different modules?
3. Regarding the ''Riemannian FFN Experts'', is the Riemannian linear transformation implemented by first projecting the manifold data into Euclidean space for computation? If so, does this render the exponential and logarithmic mappings redundant in this context?
4. What is the underlying motivation for the operation used to calculate $\hat{\kappa}_v$ in Equation (12)? Is the intention to measure node curvature by taking a weighted sum of expert curvatures? Further clarification is needed as the current formulation is difficult to interpret.
5. The manuscript suffers from significant presentation issues, as several key elements lack adequate explanation. For instance, the circles and points in Figure 1 are not defined. Furthermore, notations such as $\hat{S}$, $\kappa_v^*$ are either undefined or not explained upon their first appearance, which severely hinders the coherence and readability of the paper.

---

> ### Author Rebuttal · Authors · 2026-03-31
>
> We sincerely thank the reviewer for the time and effort in reviewing our paper. We take all comments seriously and try our best to address every raised concern. We have incorporated the corresponding revisions into the updated manuscript. Please feel free to ask any follow-up questions.
>
> ## Q1
>
> We would like to clarify the role and advantages of Gromov's $\delta$-hyperbolicity:
>
> (1) δ-hyperbolicity serves not as an exact geometry estimator, but as a weak supervisory signal and a structural prior.
>
> · It only provides a relative ordering: nodes with smaller hyperbolicity are encouraged to select lower-curvature experts. Since $\delta$-hyperbolicity and curvature are different quantities and may be noisy, we introduce the margin $\tau$ in Eq. (13) to improve robustness.
>
> · It is just one of seven graph properties used to enrich node features.
>
> (2) Since discrete Ricci curvature admits multiple definitions, we focus on the well-known Balanced Forman Ricci curvature [1]. Under this definition, both cycles with more than five nodes and interior grid edges exhibit zero curvature, making these structures difficult to distinguish. In contrast, the hyperbolicity of a cycle of length $n$ is $\lfloor n/4 \rfloor$. For sufficiently large $n$, a cycle can exhibit larger hyperbolicity than many grid structures.
>
> (3) We experimented by replacing $\delta$-hyperbolicity with mean edge Ricci curvature. **Experimental Results** (https://postimg.cc/wyLkMyqm) show that, while Ricci curvature improves performance, $\delta$-hyperbolicity yields superior results.
>
> ## Q2
> As noted by the reviewer, we employ the same curvature parameter set {-3, -2, -1, 0, 1, 2, 3} for both the Fast Riemannian Attention and Riemannian FFN modules.
> However, their GCRouters are not shared: each module has its own router to adaptively learn routing weights for the same node.
> Our motivation is that both modules aim to model the local geometry of the same node at the same timestamp. From this perspective, sharing the same curvature candidates is a natural and simple choice.
> We also appreciate the reviewer's suggestion that using different curvature settings for different modules may further enhance the model's flexibility, and we will consider it in our future work.
>
> ## Q3
> The Riemannian FFN takes the Euclidean multi-head attention output $\mathbf{O}$ as input.
> In each $\mathcal{F}^{\kappa_m}$, the input is first mapped by the exponential map to the manifold with curvature $\kappa_m$, where Möbius matvec and Möbius add are performed, and then mapped back to Euclidean space via the logarithmic map. This procedure is exactly equivalent to its Euclidean counterpart only when the curvature is zero, and is approximately equivalent when the vectors are close to the origin. We will make it more explicit in the revision.
>
> ## Q4
> We completely agree with the reviewer. Since each expert corresponds to a predefined curvature $\kappa_i$, Eq. (12),
> $\hat{\kappa}_v = \textstyle \sum _{i=1}^{M} g_i(v)\,\kappa_i$
> can be viewed as a soft estimate of node curvature induced by the routing distribution. However, it is not intended to recover the exact intrinsic curvature of a node. Instead, it is introduced for the ranking loss, so that the router can better assign each node to experts that match its local geometry. Specifically, the empirical curvature $ \kappa_v^* $ ($\delta$-hyperbolicity) acts as a coarse anchor for the node geometry: nodes with smaller $ \kappa_v^* $ are encouraged to prefer lower-curvature experts. For example, suppose node $v$ has hyperbolicity $ \kappa_v^* = 0 $ and node $u$ has $ \kappa_u^* = 0.5 $, with expert curvatures {-1, 0}. The ranking loss encourages weights such as {0.8, 0.2} for $v$ and {0.6, 0.4} for $u$, yielding routed curvatures
> $\hat{\kappa}_v = -0.8 < \hat{\kappa}_u = -0.6$. We will add an illustrative example in the manuscript.
>
> ## Q5
> (1) Clarification on Figure 1: The circles, points, and human icons all represent nodes, i.e., different people. For visual simplicity and aesthetic considerations, we used circles and points to depict some of the people.
>
> (2) Details on $\hat{\mathcal{S} }$ in Eq.(11): Here, $\mathcal{S} _i = \lbrace \mathbf{s} _i, \mathbf{s} _{i+1}, \dots, \mathbf{s} _{i+k-1}\rbrace$ denotes the sliding-window subset of routing scores starting from expert $i$ with window size $k$. Then, $\hat{\mathcal{S} }$ represents the subset whose routing scores have the largest summed value. For example, when expert number $M=5$ and $k=2$, suppose the routing scores are [0.1,0.25,0.1,0.4,0.15]. Then $\mathcal{S}_1$= [0.1, 0.25], $\mathcal{S}_2$ = [0.25, 0.1], $\mathcal{S}_3$= [0.1, 0.4], $\mathcal{S}_4$ = [0.4, 0.15], and $\hat{\mathcal{S}} $ = $\mathcal{S}_4$.
>
> (3) Definition of $\kappa_v^*$: it denotes the $\delta$-hyperbolicity of the 3-hop neighborhood of node $v$ at the current timestamp.
>
> ---
>
> [1] Understanding Over-squashing and Bottlenecks on Graphs via Curvature, ICLR 2022

---

> > ### Author Rebuttal · Reviewer_WEfN · 2026-04-03
> >
> > I will keep my score

---

> > > ### Author Response · Authors · 2026-04-04
> > >
> > > Dear Reviewer WEfN,
> > >
> > > Thank you very much for your thoughtful follow-up and for taking the time to carefully consider our rebuttal. We are very glad to know that our responses have thoroughly addressed your concerns.
> > >
> > > As dynamic graphs evolve over time, local neighborhoods may exhibit diverse and time-varying topological patterns, making it difficult for a single geometry to model them adequately. To address this challenge, we propose DyGMoCE, a dynamic graph Transformer with a mixture of curvature-aware experts, which enables each node at each timestamp to adaptively embed in a mixed-curvature space. In particular, our method introduces curvature-aware experts into both the attention and FFN modules, designs a geometrically continuous router with a ranking constraint to better align expert selection with local topology, and further develops a fast Riemannian attention module to significantly improve computational efficiency. We also conduct extensive experiments on twelve benchmark datasets covering dynamic link prediction and node classification under both transductive and inductive settings, with three negative sampling strategies. The results show that DyGMoCE achieves consistently strong performance, obtaining best or second-best results in 67/72 transductive settings and 62/72 inductive settings, while the proposed fast Riemannian attention brings an average speedup of 26.3% and a memory reduction of 52.0%. In addition, we include experiments on very large-scale datasets, further demonstrating the scalability and robustness of our method.
> > >
> > > Following your suggestions, **we have polished the manuscript as detailed below**:
> > >
> > > (1)  GCRouter is expected to promote routing behaviors that align with the intrinsic geometry encoded in the local neighborhood of each node. Specifically, we define the routed curvature of node $v$ as $\hat{\kappa}_v = \textstyle \sum _{i=1}^{M} g_i(v)\kappa_i$, serving as a soft estimate of its geometric properties.
> > > Here, $g_i(v)$ denotes the gating weight assigned to expert $E _{\kappa_i}$ associated with curvature $\kappa_i$.
> > >
> > > Let $ \kappa_{v}^{\*} $ denote the Gromov's $\delta$-hyperbolicity of the 3-hop local neighborhood of node $v$, quantifying its empirical geometry. For any two nodes $v$ and $u$ satisfying $ \kappa_{v}^{\*}<\kappa_u^{*} $, we encourage the routed curvatures to preserve this ordering by minimizing the following ranking loss:
> > > $$ \mathcal{L}_{\text{rank}}= \textstyle  \sum _{\kappa_v^{\*}<\kappa_u^{\*}} \max (0,\hat{\kappa}_v - \hat{\kappa}_u - \tau ). $$
> > > Since $\delta$-hyperbolicity characterizes the tree-likeness of a graph rather than its exact curvature, we introduce a margin hyperparameter $\tau$ to relax the ordering constraint and improve robustness.
> > >
> > > (2) Let $\mathcal{S}_i = $
> > > {
> > > $\mathbf{s}_i, \mathbf{s} _{i+1},\ldots, \mathbf{s} _{i+k-1}$
> > > }
> > > denote the routing scores of experts within $\mathcal{B}_i$. The optimal subset $\hat{\mathcal{S}}$ is the one with the maximal aggregated routing score, and $\hat{\mathcal{B}}$ denotes the corresponding expert batch.
> > >
> > >
> > > In addition, **we have rewritten the description of Figure 1** (https://postimg.cc/qgHwGBgW) and **added a notation table** (https://postimg.cc/d7VB1y87).
> > >
> > >
> > >
> > >
> > >
> > > We sincerely appreciate your careful review and constructive questions, which helped us improve both the technical clarity and the presentation of the paper. In the revised version, we will ensure the manuscript is polished. Thank you again for your valuable time and insightful feedback.
> > >
> > > Best regards,
> > >
> > > Authors

---

### Official Review · Reviewer_YNvJ · 2026-03-13

**Soundness:** 2
**Presentation:** 3
**Significance:** 2
**Originality:** 3
**Overall Recommendation:** 4
**Confidence:** 5

**Summary:**

This paper studies the issue of geometry misalignment in continuous-time dynamic graphs. The core observation is that different nodes at different timestamps can have very different local topologies. A critical point raised is that a single global geometry is too restrictive because "local" graph structure can vary over time and across regions of the graph (e.g. this can range range from more tree-like to more cyclic or flatter neighborhoods). To address this core issue (along with accelerating Riemannian operations), the paper proposes DyGMoCE, a dynamic graph Transformer with a mixture of curvature-aware experts used inside both the attention and FFN layers. The proposed model includes a routing mechanism that selects a contiguous block of experts on an ordered curvature axis, a ranking loss that aligns routed curvature with empirical local geometry estimated via neighborhood hyperbolicity, and a fast Riemannian attention implementation based on a polar-coordinate reformulation.

**Compliance With Llm Reviewing Policy:**

Affirmed.

**Final Justification:**

The paper's core design -- curvature-ordered experts with contiguous routing and a fast Riemannian attention reformulation -- is technically sound (I verified the derivations in Eqs. 16-17) and gives the method genuine identity beyond generic MoE. My original score of 2 was driven by three critical gaps: no formal equivalence statement (Q1), no evidence that topological diversity actually exists at the per-node level (Q3), and no parameter-matched Euclidean ablation (Q8). The rebuttal addresses all three directionally -- providing a theorem statement, CV analysis against random-graph baselines showing real cross-node heterogeneity, and a 7-expert Euclidean comparison. I raise my score to 4 accordingly.

**Key Questions For Authors:**

1****. The paper says (at multiple places) Fast RA is "mathematically equivalent" to conventional Riemannian attention, but never formalizes this as a theorem or proposition. Equivalence under what conditions? The polar decomposition divides by the norm, so what happens when Q_i or V_j is near zero? Does this hold in finite precision? A formal statement with explicit assumptions is needed for a claim this central.

2**. What is the preprocessing time for computing delta-hyperbolicity across all nodes and timestamps on the larger datasets (LastFM, Social Evo., Contact, Flights)? This is O(n^4) per subgraph even with BCCM pruning (which is quite a bit), and is never reported.

3****: The paper's core claim is that local topology varies both across nodes and over time (intuitively, surely makes sense), yet the only "empirical evidence" is two numbers per dataset (initial and final average hyperbolicity in Table 5). Can you provide a more thorough data analysis -- at minimum, distributions of per-node delta-hyperbolicity at multiple timestamps for the larger datasets, and time-series of the mean and variance? Crucially, I need to see whether different nodes genuinely exhibit different geometries at the same timestamp (cross-node heterogeneity), and whether individual nodes' geometries actually change over time (temporal evolution). If the shift from 0.019 to 1.850 on MOOC is just a global uniform drift, then per-node adaptive curvature is solving a problem that doesn't exist. This analysis would either validate the paper's foundational premise or reveal that a simpler time-varying global curvature would suffice.

4***. Can the authors provide evidence that the curvature-aware routing is actually learning to specialize by geometry? For instance, do nodes with low delta-hyperbolicity consistently route to hyperbolic experts post-training? A visualization of routing patterns vs. empirical hyperbolicity would significantly strengthen the paper's narrative. Without this, I'm not fully convinced the routing isn't just learning arbitrary expert preferences that happen to work.

5***. Why does DyGMoCE perform so poorly on UN Vote dataset in the inductive setting? DyGMoCE scores 52.49 AUC-ROC on UN Vote inductively (Table 8, rnd) while TIDFormer reaches 61.59. The paper highlights UN Vote as a curvature-diversity success story transductively. Same question for the ~7 AP gap on Flights (hist) and rank-4 on Contact (rnd) -- none discussed.

6**. Why were CAT and GraphMoRE (discussed in related work as the most closely related MoE-based curvature methods) not included in the experimental comparison? Even if they target static graphs, comparing them on the same dynamic graph benchmarks (or at least on static snapshots) would help isolate the contribution of DyGMoCE's specific design choices.

7*. On datasets where DyGMoCE's improvement is marginal (e.g., Wikipedia, Reddit, Contact), does the model degenerate to effectively selecting a single curvature? In other words, is the routing learning that mixed-curvature isn't needed here, or is it still selecting diverse experts but not gaining from it? I would need some clarification here to be convinced that the added complexity is justified.

8****. True euclidean only ablation is missing. The ablations remove Top-k, ranking, and geometric encoding, but they do not include a "parameter-matched" Euclidean MoE Transformer or Euclidean multi-expert baseline with the same routing machinery. Without this, it is very difficult to be sure that the gains are stemming in from geometric inductive biases.

9**. In Eq. (13), the ranking loss is driven by local delta-hyperbolicity, but Appendix B.3 defines delta as a tree-likeness statistic over 3-hop neighborhoods rather than a curvature estimate. Why should the ordering induced by delta be monotone with the ordered curvature grid kappa_i used by GCRouter?

**Limitations:**

The authors discuss some limitations like expert redundancy in Appendix A, which is appreciated. However, the more pressing limitations are not adequately addressed (see the limitations and key questions mentioned above).

**Strengths And Weaknesses:**

### Strengths

The motivation is intuitive and sensible: continuous-time dynamic graphs can exhibit locally different structural regimes, and a single fixed geometry may be too restrictive. The proposed design is also more thoughtful than a generic MoE adaptation. In particular, the use of curvature-ordered experts, contiguous routing on the curvature axis, and a faster reformulation of Riemannian attention gives the method a clear identity. However, there are several limitations which are hindering my confidence in the paper.

### Weaknesses

In my opinion, the paper’s conceptual narrative is stronger than the evidence currently provided for it, I discuss this in "key questions". The paper argues for per-node, time-varying geometric heterogeneity and geometry-aware expert specialization, but neither of these is yet validated as directly as I would like. Relatedly, some of the strongest technical claims, especially around Fast RA, are stated more strongly than they are formalized, and the experimental section is missing an important control needed to isolate the effect of geometric inductive bias from generic MoE capacity. See Q1–Q9 for the main issues on which I would like clarification from the authors.

---

> ### Author Rebuttal · Authors · 2026-03-31
>
> We sincerely thank the reviewer for the thoughtful feedback. We have carefully revised the manuscript to address the concerns and would be happy to clarify any further questions.
>
> ## W1
> **Theorem 1.** Let $Q_i$, $K_j$,  $V_j$ be query, key, and value vectors. Fast Riemannian attention is equivalent to Riemannian attention if the following conditions hold:
>
> *a)* $Q_i$, $K_j$, and $V_j$ lie on the $\kappa$-stereographic manifold $\mathcal{M}_{\kappa} $= {$ \mathbf{x}  \mid -\kappa \|\mathbf{x}\|^2 < 1$ }, whose Riemannian metric is conformal to the Euclidean metric;
>
> *b)* For any zero vectors, define the corresponding geodesic radial coordinate $\rho=0$, and the direction **$u$** as a unit vector (or arbitrary).
>
> **Cases where $Q_i$ or $V_j$ equal zero**
>
> When $Q_i = 0$, point $A$ coincides with point $O$ in Figure 3. The geodesic distance $ \mathcal{M}_{\kappa}(Q_i, K_j) = \cos_k^{-1}(\cos_k(\rho_j^K))$ in Eq.(16) equals the length of $OB$, i.e., $\rho_j^K$.
>
> When $V_j = 0$, we have $\rho_j^V=0$. Therefore, $ w_{ij}\rho_j^V\mathbf{u}_j^V =0$.
>
> For zero vectors, $\mathbf{u}$ is not involved in the computation and can be set arbitrarily.
>
> **Behavior under finite precision**
>
> We use torch.norm(x, dim=-1).clamp_min(1e-15) in the code; see Lines 638–660 of DyGMoCE-main/models/DyGMoCE.py.
>
> ## W2
>
> BCCM pruning does not change the worst-case complexity of computing hyperbolicity for each 3-hop subgraph. In dynamic graphs, however, repeated historical links often do not enlarge local subgraphs, stale edges are discarded, and subgraphs can be processed in parallel. **Preprocessing runtimes**: https://postimg.cc/d7VB1y81.
>
>
>
> ## W3
>
>
> We use the mean $\mu$, variance $\sigma$, and coefficient of variation ($CV=\sigma/\mu$) to quantify hyperbolicity variation, where a larger $CV$ indicates stronger relative fluctuation. To verify cross-node geometric heterogeneity at the same timestamp, **we examine every 10% of the full timeline**  https://postimg.cc/DSgxhbj3, https://postimg.cc/FYZTvJW4. Compared with Barabási-Albert and Erdős–Rényi graphs of similar size and degree, our real datasets show clearly larger $CV$s. To verify temporal evolution, **we compute per-node $CV$ over active timestamps**, and observe clear variation.  https://postimg.cc/NLdGKNKs, https://postimg.cc/w3qTTKjv
>
>
> ## W4
>
> First, in Section 4.2, GCRouter uses ranking constraints as weak supervision to explicitly encourage nodes with low $\delta$-hyperbolicity to be routed to experts with relatively smaller curvature, rather than *learning arbitrary expert preferences*.
>
> Second, **we provide more direct evidence through a “routing patterns vs. empirical hyperbolicity” analysis**, which shows that the router assigns experts to nodes in a geometry-aware manner. https://postimg.cc/BLwy01v9, https://postimg.cc/9RfSKBcj, https://postimg.cc/7GLcdNqZ
>
>
> ## W5
>
> The inductive setting requires predicting edges involving new nodes. In UNVote, the new-edge ratio is extremely low. Moreover, subgraphs with fewer than 4 nodes cannot support hyperbolicity computation, which may cause degenerate geometry estimation. Historical NSS samples negatives from previously observed but currently inactive edges, a setting that may favor RepeatMixer on Flights due to its explicit modeling of repeated interactions. On the Contact dataset, DyGMoCE ranks fourth but is only 0.21 behind the best method, indicating a minor ranking fluctuation. Overall, DyGMoCE achieves best or second-best performance in 67/72 transductive and 62/72 inductive settings, confirming its effectiveness.
>
>
>
> ## W6
>
> CAT and GraphMoRE are designed for static graphs. Adapting them to dynamic link prediction, even via snapshots, requires additional temporal designs. Following your suggestion, we equipped both with the same time module as ours. https://postimg.cc/2qSw6Z41
>
>
>
> ## W7
>
> When the hyperbolicity shift is minimal, the GCRouter naturally converges toward the optimal experts. Although our method may not yield substantial gains at all times, it does not introduce significant noise or degrade performance.
>
>
> ## W8
>
> In Section 5.5, we already include an only-$\mathbb{E}$ Euclidean MoE Transformer with three Euclidean experts and the same router, which consistently underperforms the mixed-curvature model. We further add a parameter-matched Euclidean multi-expert baseline with seven Euclidean experts in both attention and FFN. https://postimg.cc/5H230CwY.
>
>
>
> ## W9
> We do not assume a strict monotonic relation between empirical $\delta$-hyperbolicity $\kappa_v^*$ and routed curvature $\hat{\kappa}_v$. In Eq. (13), $\delta$- hyperbolicity serves only as a weak supervisory signal: nodes with smaller hyperbolicity are encouraged to prefer lower-curvature experts. Since hyperbolicity and curvature are different quantities and need not align exactly, we introduce the margin $\tau$ to relax the ordering constraint. The final expert assignment is learned adaptively by the router, and our ablations support this design.

---

> > ### Author Rebuttal · Reviewer_YNvJ · 2026-04-03
> >
> > I appreciate the hard work put into the rebuttal. Most of my concerns have been resolved so I am increasing my score!

---

> > > ### Author Response · Authors · 2026-04-04
> > >
> > > Dear Reviewer YNvJ,
> > >
> > > Thank you very much for your positive follow-up and for increasing your score after considering our rebuttal. We are very glad to know that our responses have adequately addressed your concerns.
> > >
> > > As dynamic graphs evolve over time, local neighborhoods may exhibit diverse and time-varying topological patterns, making it difficult for a single geometry to model them adequately. To address this challenge, we propose DyGMoCE, a dynamic graph Transformer with a mixture of curvature-aware experts, which enables each node at each timestamp to adaptively embed in a mixed-curvature space. In particular, our method introduces curvature-aware experts into both the attention and FFN modules, designs a geometrically continuous router with a ranking constraint to better align expert selection with local topology, and further develops a fast Riemannian attention module to significantly improve computational efficiency.
> > >
> > > We sincerely appreciate your careful reading, constructive questions, and encouraging evaluation. Your thoughtful review helped us further improve the clarity and completeness of the paper. In the revised version, we will carefully incorporate the main clarifications and additional analyses from the rebuttal into the manuscript. Thank you again for your valuable time, insightful comments, and supportive evaluation.
> > >
> > > Best regards,
> > >
> > > Authors

---

### Decision · Program_Chairs · 2026-04-30

**Decision:**

Accept (regular)

**Comment:**

After the rebuttal, authors have solved all concerns. The reviewer WEfN still maintained a negative opinion. However, he did not present a sufficiently strong argument. Overall, the evaluation of this paper is positive, so I decide to accept it.